# Trade-off Between Efficiency and Consistency for Removal-based Explanations

**Yifan Zhang**[*1]    **Haowei He**[*1]    **Zhiquan Tan**[2]    **Yang Yuan**[1,3,4†]

[1]IIIS, Tsinghua University
[2]Department of Math, Tsinghua Univesity
[3]Shanghai Artificial Intelligence Laboratory
[4]Shanghai Qizhi Institute
{zhangyif21,hhw19,tanzq21}@mails.tsinghua.edu.cn
yuanyang@tsinghua.edu.cn

## Abstract

In the current landscape of explanation methodologies, most predominant approaches, such as SHAP and LIME, employ removal-based techniques to evaluate the impact of individual features by simulating various scenarios with specific features omitted. Nonetheless, these methods primarily emphasize efficiency in the original context, often resulting in general inconsistencies. In this paper, we demonstrate that such inconsistency is an inherent aspect of these approaches by establishing the Impossible Trinity Theorem, which posits that interpretability, efficiency, and consistency cannot hold simultaneously. Recognizing that the attainment of an ideal explanation remains elusive, we propose the utilization of interpretation error as a metric to gauge inefficiencies and inconsistencies. To this end, we present two novel algorithms founded on the standard polynomial basis, aimed at minimizing interpretation error. Our empirical findings indicate that the proposed methods achieve a substantial reduction in interpretation error, up to 31.8 times lower when compared to alternative techniques[‡].

## 1 Introduction

Most existing explanation approaches are removal-based [18], which involve the sequential process of eliminating certain input features, examining the subsequent alterations in the model's behavior, and ascertaining each feature's impact through observation. However, most of these methods [60, 38] primarily focus on the original input with all features, often yielding inconsistent outcomes in alternative scenarios. Consequently, the resulting interpretations may not explain the network's behavior consistently even in a small neighborhood of the input, as demonstrated in Figure 1.

Inconsistency is a non-trivial concern. Imagine a doctor treating a diabetic patient with the help of an AI system. The patient has features A, B, and C, representing three positive signals from various tests. The AI recommends administering 4 units of insulin with the following explanation: A, B, and C have weights 1, 1, and 2, respectively, amounting to 4 units in total. The doctor might then ask the AI: *what if* the patient only has A and B, but not C? One might expect the answer to be close to 2, as A+B has a weight of 2. However, the network, being highly non-linear, might output a different suggestion like 3 units, explaining that both A and B have a weight of 1.5. Such inconsistent

---

[*]Equal Contribution.

[†]Corresponding Author

[‡]Code is available at https://github.com/trusty-ai/efficient-consistent-explanations.

37th Conference on Neural Information Processing Systems (NeurIPS 2023).

behaviors can significantly reduce the doctor's confidence in the interpretations, limiting the practical value of the AI system.

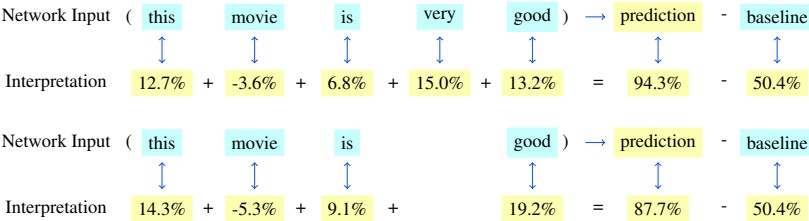

Figure 1: Interpretations generated by SHAP on movie review.

Consistency (see Definition 2.2) is certainly not the only objective for interpretability. Being equally important, efficiency is a commonly used axiom in the attribution methods [72, 22, 59], also called local accuracy [38] or completeness [60], stating that the model's output should be equal to the network's output for the given input (see Definition 2.3). Naturally, one may ask the following question:

> **Q1: Can we generate an interpreting model that is both consistent and efficient?**

Unfortunately, this is generally unattainable. We have proved the following theorem in Section 2:

**Theorem 1**(Impossible trinity, informal version). *For post-hoc interpreting models, interpretability, efficiency and consistency cannot hold simultaneously.*

A few examples following Theorem 1:
  (a) Most attribution methods are interpretable and efficient, but not consistent.
  (b) The original (deep) network is consistent and efficient, but not interpretable.
  (c) If one model is interpretable and consistent, it cannot be efficient.

However, consistency is necessary for many scenarios, leading to the follow-up question:

> **Q2: For consistent interpreting models, can they be approximately efficient?**

The answer depends on the definition of "approximately efficient". We introduce a new notion called *truthfulness*, which serves as a natural relaxation of efficiency, or partial efficiency. We divide the functional space of a network $f$ into two subspaces: the readable part and the unreadable part. We call the interpreting model $g$ truthful if it can accurately represent the readable part of $f$, denoted as $V$ (see Definition 2.4). The unreadable part is not merely a "fitting error"; it truthfully represents the higher-order non-linearities in the network that our interpretation model $g$, even at its best, cannot cover. In short, what $g$ conveys is true, though it may not encompass the entire truth. Due to Theorem 1, **this is essentially the best that consistent algorithms can achieve.**

Truthfulness is a parameterized notion, depending on the choice of the readable subspace. While there are theoretically infinite possible choices, we follow previous researchers on interpretability with non-linearities [61, 41, 63], using the basis that induces interpretable terms like $x_i x_j$ or $x_i x_j x_k$. These capture higher-order correlations and are easy to understand. The resulting subspace has the standard polynomial basis (or equivalently, the Fourier basis).

When truthfulness is parameterized with the polynomial basis, designing consistent and truthful interpreting models is equivalent to learning the Fourier spectrum (see Algorithm 1 and Lemma 3.4). However, exact consistency is not always necessary for most real-world applications, as approximate consistency is usually sufficient. We formally use the number of different interpreting models (in log scale) as a metric for inconsistency. Our last question is:

> **Q3: When a little inconsistency is allowed, can we get better interpreting models?**

We affirmatively answer this question in Section 4 by introducing a new algorithm called Harmonica-local. We then apply it multiple times to develop Harmonica-anchor and Harmonica-anchor-constrained, which offer smaller interpretation errors with a small degree of inconsistency.

In this paper, we focus on removal-based explanations [18], which implies that $f$ and $g$ are Boolean functions. We remark that empirically most networks are not Boolean functions, i.e., the input variables are real valued. However, for every explanation algorithm (including LIME [47] and SHAP [38]) in the removal-based framework, the input $x$ is not modified to an arbitrary real value. Instead, $x$ is fixed, and the algorithm only retain or remove each feature of $x$ for generating the explanations. When $x$ is fixed with $n$ features, it becomes natural to use Boolean functions to represent both the network $f$ and the interpreting model $g$ for their outputs in all $2^n$ feature-removal scenarios. In other words, the algorithms in the framework only consider a fixed $x$ each time, and given this $x$, there are only $2^n$ possible outcomes, even for the models with real inputs. Therefore, treating $f$ and $g$ as Boolean functions is not a simplification, but an accurate characterization of the removal-based framework (see Figure 1 in [18] for an illustration).

Our new algorithms, including Harmonica-local, Harmonica-anchor, and Harmonica-anchor-constrained, are all based on the Harmonica algorithm from Boolean functional analysis [27], which has rigorous theoretical guarantees on recovery performance and sampling complexities. In Section 5, we demonstrate that on datasets like IMDb and ImageNet, our algorithms achieve up to 31.8x lower interpretation error compared with other methods.

In summary, our contributions are:

- We prove the impossible trinity theorem for removal-based explanations, demonstrating that interpretable algorithms cannot be consistent and efficient simultaneously.
- When a small inconsistency is allowed, we propose new algorithms using Harmonica-local and empirically demonstrate that these algorithms achieve significantly lower interpretation errors compared with other methods.
- For interpretable algorithms that are consistent but not efficient, we introduce a new notion called truthfulness, which can be regarded as partial efficiency. Due to the impossible trinity theorem, this is the best achievable outcome when consistency is required.

## 2 Our Framework on Interpretability

We consider a Hilbert space $\mathcal{H}$ equipped with inner product $\langle \cdot, \cdot \rangle$, and induced norm $\| \cdot \|$. We denote the input space by $\mathcal{X}$, the output space by $\mathcal{Y}$, which means $\mathcal{H} \subseteq \mathcal{X} \to \mathcal{Y}$. We use $\mathcal{G} \subset \mathcal{H}$ to denote the set of interpretable functions, and $\mathcal{F} \subset \mathcal{H}$ to denote the set of machine learning models that need interpretation. In this paper, if not mentioned otherwise we focus on models that are not self-interpretable, i.e., $f \in \mathcal{F} \setminus \mathcal{G}$.

**Definition 2.1** (Interpretable and Interpretation Algorithm). We call A model $g$ is *interpretable*, if $g \in \mathcal{G}$. An *interpretation algorithm* $\mathcal{A}$ takes $f \in \mathcal{H}, x \in \mathcal{X}$ as inputs, and outputs $\mathcal{A}(f, x) \in \mathcal{G}$ for interpreting $f$ on $x$.

As we mentioned previously, for many interdisciplinary fields, the interpretation algorithm should be consistent.

**Definition 2.2** (Consistent). Given $f \in \mathcal{H}$, an interpretation algorithm $\mathcal{A}$ is *consistent* with respect to $f$, if $\mathcal{A}(f, x)$ remains the same (function) for every $x \in \mathcal{X}$.

Efficiency is an important property of the attribution methods.

**Definition 2.3** (Efficient). A model $g \in \mathcal{H}$ is *efficient* with respect to $f \in \mathcal{F}$ on $x \in \mathcal{X}$, if $g(x) = f(x)$.

The following theorem states that one cannot expect to achieve the best of all three worlds.

**Theorem** 1 (**Impossible Trinity for Removal-based Explanations**). For any interpretation algorithm $\mathcal{A}$ and function sets $\mathcal{G} \subset \mathcal{F} \subseteq \mathcal{H}$, there exists $f \in \mathcal{F}$ such that with respect to $f$, either $\mathcal{A}$ is not consistent, or $\mathcal{A}(f, x)$ is not efficient on $x$ for some $x \in \mathcal{X}$.

*Proof.* Please refer to Appendix B for all the proofs. □

Theorem 1 says efficiency is too restrictive for consistent interpretations. However, being inefficient does not mean the interpretation is wrong, it can still be truthful. Recall a subspace $V \subset \mathcal{H}$ is *closed* if whenever $\{f_n\} \subset V$ converges to some $f \in \mathcal{H}$, then $f \in V$. We have:

**Definition 2.4** (Truthful gap and truthful). Given a closed subspace $V \subseteq \mathcal{H}$, $g \in \mathcal{G} \subseteq V$ and $f \in \mathcal{F}$, the *truthful gap* of $g$ to $f$ for $V$ is:

$$\mathbb{T}_V(f,g) \triangleq \|f - g\|^2 - \inf_{v \in V} \|f - v\|^2. \tag{1}$$

When $\mathbb{T}_V(f,g) = 0$, we say $g$ is *truthful* for subspace $V$ with respect to $f$, and we know (see e.g. Lemma 4.1 in [56]) $\forall v \in V, \langle f - g, v \rangle = 0$.

Truthfulness means $g$ fully captures the information in the subspace $V$ of $f$, therefore it can be seen as a natural relaxation of efficiency. To characterize the interpretation quality, we introduce the following notion.

**Definition 2.5** (Interpretation error). Given functions $f, g \in \mathcal{X} \to \mathcal{Y}$, the interpretation error between $f$ and $g$ with respect to measure $\mu$ is

$$\mathbb{I}_{p,\mu}(f,g) \triangleq \left( \int_{\mathcal{X}} |f(x) - g(x)|^p d\mu(x) \right)^{1/p}. \tag{2}$$

Notice that interpretation error is only a *loss function* that measures the quality of the interpretation, instead of a metric in $\ell_p$ space. Therefore, $\mu$ can be a non-uniform weight distribution following the data distribution. If $\mu$ is uniform distribution over $\mathcal{X}$, we abbreviate $\mathbb{I}_{p,\mu}(f,g)$ as $\mathbb{I}_p(f,g)$. For real-world applications, interpreting the model over the whole $\mathcal{X}$ is unnecessary, so $\mu$ is usually defined as a uniform distribution on the neighborhood of input $x$ (under a certain metric), in which case we denote the distribution as $\mathcal{N}_x$.

## 3 Applying Our Framework to Removal-based Explanations

Now we focus on interpreting removal-based explanations [38, 18]. Removal-based explanations are post-hoc, which means they are generated based on a target network for a fixed input, by removing features from that input. There are three choices affecting the removal-based explanations: how features are removed, what behavior is analyzed after feature removal, and how to summarize the feature influence. For example, SHAP [38] considers all possible subsets of the features, and analyzes how holding out different features affects functional value, and finally summarizes the differences based on the Shapley value calculation.

Therefore, removal-based explanations can be represented as Boolean functions, as feature subset $S \subseteq [n]$ can be represented as Boolean input: $f \in \{-1, 1\}^n \to \mathbb{R}$, where $-1$ means the specific feature $i \in S$ is removed, $1$ means the specific feature is retained. We use $-1/1$ instead of $0/1$ to represent the binary variables for ease of exposition using the Fourier basis.

### 3.1 Fourier Basis and Truthful Gap

Fourier analysis is a handy tool for analyzing Boolean functions. Due to the space limit, we defer the more comprehensive introduction on Fourier analysis for Boolean functions to Appendix A, and only present the necessary notions here.

**Definition 3.1** (Fourier basis). For any subset of variables $S \subseteq [n]$, we define the corresponding Fourier basis as $\chi_S(x) \triangleq \Pi_{i \in S} x_i \in \{-1, 1\}^n \to \{-1, 1\}$.

The Fourier basis is also called polynomial basis in the literature. It is a complete orthonormal basis for Boolean functions, under the uniform distribution on $\{-1, 1\}^n$. We remark that this uniform distribution is used for theoretical analysis and algorithm design, and is different from the measure $\mu$ for interpretation quality assessment in Definition 2.5.

**Definition 3.2** (Fourier expansion). Any Boolean function $f \in \{-1, 1\}^n \to \mathbb{R}$ can be expanded as

$$f(x) = \sum_{S \subseteq [n]} \hat{f}_S \chi_S(x),$$

where $\hat{f}_S = \langle f, \chi_S \rangle$ is the Fourier coefficient on $S$.

Now we define the notion of $C$-Readable function.

**Definition 3.3** ($C$-Readable function). Given a set of Fourier bases $C$, a function $f$ is $C$-readable if it is supported on $C$. That is, for any $\chi_S \notin C$, $\langle f, \chi_S \rangle = 0$. Denote the corresponding subspace as $V_C$.

The Readable notion is parameterized with $C$, because it may differ case by case. If we set $C$ to be all the single variable bases, only linear functions are readable; if we set $C$ to be all the bases with the degree at most 2, functions with pairwise interactions are also readable. Moreover, if we further add one higher order term to $C$, e.g., $\chi_{\{x_1,x_2,x_3,x_4\}}$, it means we can also reason about the factor $x_1 x_2 x_3 x_4$ in the interpretation, which might be an important empirical factor that people can easily understand. Starting from the bases set $C$, we have the following formula for computing the truthful gap.

**Lemma 3.4** (Truthful gap for Boolean functions). *Given a set of Fourier bases $C$, two functions $f, g \in \{-1, 1\}^n \to \mathbb{R}$, the truthful gap of $g$ to $f$ for $C$ is*

$$\mathbb{T}_{V_C}(f, g) = \sum_{\chi_S \in C} \langle f - g, \chi_S \rangle^2. \tag{3}$$

With the previous definitions, it becomes clear that finding a truthful interpretation $g$ is equivalent to accurately learning a Boolean function with respect to the readable bases set $C$. Intuitively, it means we want to find algorithms that can compute the coefficients for the bases in $C$. In other words, we want to find the importance of the bases like $x_1, x_2 x_5, x_2 x_6 x_7$, etc.

## 3.2 Representative Algorithms

Applying the impossible trinity theorem to removal-based explanations, there are two notable algorithms on the extremes.

**Shapley values: efficient but not consistent.** Let $N = \{1, 2, ..., n\}$ represent a set of $n$ players, and let $v : 2^N \to \mathbb{R}$ be a characteristic function that assigns a real value to each coalition of players. The Shapley value of player $i$ is given by:

$$\phi_i(v) = \sum_{S \subseteq N \setminus i} \frac{(n - |S| - 1)! \cdot |S|!}{n!} [v(S \cup i) - v(S)], \tag{4}$$

where $|S|$ is the number of players in coalition $S$, and the sum is taken over all possible coalitions $S$ that do not include player $i$. One of the most important properties of Shapley values is *efficiency*, i.e. $\sum_{i \in N} \phi_i(v) = v(N) - v(\emptyset)$ (or denoting explanation function $g \triangleq \sum_{i \in N} \phi_i(v) + v(\emptyset)$). As Shapley value based explanations do **not** focus on consistency, its explanation is efficient only for the original input, but not for other scenarios when certain features are removed.

**Harmonica: consistent but not efficient.** What can we do if we want to address consistency? As mentioned in Definition 2.4, we do not expect the algorithm to be efficient, but it can still be truthful with respect to a subspace $V$, which is naturally represented with the polynomial basis. Therefore, in order to learn truthful explanations for given subspace $V$, it is natural to consider LASSO regression over the coefficients on the polynomial basis. Harmonica (Algorithm 1) fulfills this requirement [27] and has superior complexity shown in Theorem 2.

---

**Algorithm 1:** Harmonica [27]

1. Given uniformly randomly sampled $x_1, \cdots, x_T$, evaluate them on $f$: $\{f(x_1), ...., f(x_T)\}$.
2. Solve the following regularized regression problem.

$$\underset{\alpha \in \mathbb{R}^{|C|}}{\operatorname{argmin}} \left\{ \sum_{i=1}^{T} \left( \sum_{S, \chi_S \in C} \alpha_S \chi_S(x_i) - f(x_i) \right)^2 + \lambda \|\alpha\|_1 \right\} \tag{5}$$

3. Output the polynomial $g(x) = \sum_{S, \chi_S \in C} \alpha_S \chi_S(x)$.

---

**_Theorem_ 2** (Complexity of Harmonica). Given $f \in \{-1, 1\}^n \to \mathbb{R}$, a $(\epsilon/4, s, C)$-bounded function, Algorithm 1 finds a function $g$ with interpretation error at most $\epsilon$ in time $O((T \log \frac{1}{\epsilon} + |C|/\epsilon) \cdot |C|)$ and sample complexity $T = \tilde{O}(s^2/\epsilon \cdot \log |C|)$.

We defer detailed descriptions and theoretical guarantees of these algorithms to Appendix B.1, C and D, discussion on the comparison of our algorithms with the existing algorithms to Appendix E.

## 3.3 Quantifying Inconsistency

All the Fourier coefficients together are called the Fourier spectrum of $f$. We can define the following distance between two functions based on their spectrums as the quantitative measure of the inconsistency between different explanations.

**Definition 3.5** (Spectrum (Fourier) distance for Boolean functions). Given two interpretations $g(\cdot), h(\cdot) \in \{-1, 1\}^n \to \mathbb{R}$, we define the *spectrum p-distance* between them as:

$$\mathbb{D}_p\left(g(\cdot), h(\cdot)\right) \triangleq \hat{\|}g - h\hat{\|}_p = \left( \sum_{S \subseteq [n]} \left| \hat{g}_S - \hat{h}_S \right|^p \right)^{1/p}. \tag{6}$$

As there may be many candidates $p$ for evaluating interpretation error and spectrum distance. In the following, we will investigate the most natural choice for $p$. As the Fourier expansion uniquely determined a Boolean function. We would like the distance between their expansion to have the same tendency as the interpretation error, thus the spectrum can encode enough information to reflect the accuracy of the interpretation. We shall first discuss the most "strict" case, i.e. the $L_0$ norm.

Denote the support of an Boolean function $f$ as $\mathrm{supp}(f) \triangleq \{x \in \{-1, 1\}^n \mid f(x) \neq 0\}$. The support of Fourier coefficients (Fourier spectrum), is defined by $\mathrm{supp}(\hat{f}) \triangleq \{S \subseteq [n] \mid \hat{f}(S) \neq 0\}$. Based on Proposition C.1, we have the following *uncertainty principle for removal-based explanations*:

***Theorem* 3** (Uncertainty Principle for Removal-based Explanations). Assume $f, g \in \{-1, 1\}^n \to \mathbb{R}$ and $f \neq g$. When evaluated under $L_0$ norm, the interpretation error and spectrum distance (inconsistency value) have the following uncertainty principle (can be seen as a quantitative version of Impossible Trinity Theorem 1):

$$\mathbb{I}_0(f, g)\mathbb{D}_0(f, g) \geq 1. \tag{7}$$

We can see that the $L_0$ norm is too strict for evaluating the differences between interpretations (inconsistency). Fortunately, the following proposition shows that the $L_2$ relaxed version is much better, or to say, natural, due to Parseval's identity.

**Proposition 3.6** ($L_2$ norm is natural). *When taking $\mu$ as a uniform distribution, for $f, g \in \{-1, 1\}^n \to \mathbb{R}$, we have $\mathbb{I}_2(f, g) = \mathbb{D}_2(f, g)$.*

Now we present a theorem on the trade-off between efficiency and consistency, quantified by total interpretation error and total inconsistency value (spectrum distance).

***Theorem* 4** (Trade-off between Efficiency and Consistency). Given a function $f : \{-1, 1\}^n \to \mathbb{R}$ and a set of $N$ interpretable functions $\{g_1, g_2, \ldots, g_N\}$ defined on disjointed sets $\mathcal{X}_i$ ($\mathcal{X} = \bigcup \mathcal{X}_i$), each $g_i : \{-1, 1\}^n \to \mathbb{R}$, let $\bar{g}$ denote the average (spectrum) of $g_i$ across the space $\mathcal{X} = \{-1, 1\}^n$. Define (we use $L_2$ norm as default):

$$\mathbb{I}_{\mathrm{total}} = \sum_{i=1}^{N} \|f|_{x \in \mathcal{X}_i} - g_i|_{x \in \mathcal{X}_i}\|_2,$$

$$\mathbb{D}_{\mathrm{total}} = \sum_{i=1}^{N} \hat{\|}g_i|_{x \in \mathcal{X}_i} - \bar{g}|_{x \in \mathcal{X}_i}\hat{\|}_2,$$

Then, we have the following inequality:

$$\mathbb{I}_{\mathrm{total}} + \mathbb{D}_{\mathrm{total}} \geq \sum_{i=1}^{N} \|f|_{x \in \mathcal{X}_i} - \bar{g}|_{x \in \mathcal{X}_i}\|_2.$$

In addition, denote $g^{\dagger}$ as the globally consistent interpretation $g^{\dagger}$ defined on $\{-1, 1\}^n$ with minimal interpretation error (can be obtained by Harmonica), we have:

$$\sum_{i=1}^{N} \|f|_{x \in \mathcal{X}_i} - \bar{g}|_{x \in \mathcal{X}_i}\|_2 \geq \sum_{i=1}^{N} \|f|_{x \in \mathcal{X}_i} - g^{\dagger}|_{x \in \mathcal{X}_i}\|_2 = \mathbb{I}_2(f, g^{\dagger}) = \mathbb{D}_2(f, g^{\dagger}).$$

**Algorithm 2:** Harmonica-anchor

---

**Input:** anchor number $k$, model $f$, a distance metric function $d(\cdot, \cdot) : (\chi_S, \chi_S) \to \mathbb{R}$ which calculates distance between two anchors (e.g., $L_p$ norm or Hamming distance), sampling number $T$, regularization coefficient $\lambda_1$.

**Output:** interpretation models $g_i$ with coefficients $\alpha_i \in \mathbb{R}^{|C|}$, $i = 1, 2, 3, ..., k$.

1: Fix random bases $b_i, i = 1, 2, 3, ..., k$ as anchors.
2: Randomly sample $T$ bases $b_1, b_2, ..., b_T$ and accordingly, we calculate $f(x_1), f(x_2), ..., f(x_T)$.
3: Assign each basis to the anchor with minimal distance and index using
$d_i = \min\{\arg\min_j d(b_i, b_j^x)\}, i = 1, 2, 3, ...T$ and $j = 1, 2, 3, ..., k$.
4: **for** $i = 1, 2, 3, ..., k$ **do**
5:    Solve the following regularized regression problem:

$$\operatorname*{argmin}_{\alpha_i \in \mathbb{R}^{|C|}}\left\{ \sum_{n=1}^{T} \left( \mathbb{I}(i = d_n)\Big( \sum_{S,\chi_S \in C} \alpha_{i,S}\chi_S(x_n) - f(x_n) \Big) \right)^2 + \lambda\|\alpha_i\|_1 \right\}. \qquad (8)$$

6: **end for**

---

This theorem illuminates the inherent trade-offs in designing interpretable models. Specifically, to reduce $\mathbb{I}_{\text{total}}$, one would typically need to increase $\mathbb{D}_{\text{total}}$ unless $f$ itself is close to an interpretable model $\bar{g}$. Theorem 4 can also be seen as a quantitative version of the Impossible Trinity Theorem 1 for removal-based explanations.

Now we established the lower bound of $\mathbb{I}_{\text{total}} + \mathbb{D}_{\text{total}}$ by $\mathbb{I}_2(f, g^\dagger)$ (the same as $\mathbb{D}_2(f, g^\dagger)$), showing the trade-off between efficiency and consistency, and also bridging the locally consistent interpretations $g_i$ and globally consistent interpretation $g^\dagger$.

## 4 Trade-off Between Efficiency and Consistency

Harmonica recovers functions across the entire function space. However, in our setting, if $\mathcal{N}_x$ is small, it is sufficient to recover a small neighborhood of the function. This inspires us to apply Harmonica to a local space instead of the entire space. By doing so, we obtain more concentrated samples in the local neighborhood. Consequently, minimizing the interpretation error in this neighborhood becomes more manageable. We refer to Harmonica with samples in the local neighborhood as Harmonica-local.

If we relax the consistency requirement, meaning that users are willing to accept minor inconsistencies in interpretations when a slight modification is made to the input, we can achieve smaller interpretation error with Harmonica-anchor, as shown in Algorithm 2. Intuitively, Harmonic-anchor applies multiple Harmonica-local algorithms to different subspaces of the input. Specifically, for a given neighborhood region $\mathcal{N}x$, we now randomly select $k_x$ bases $b_{x,i}$, $(i = 1, 2, 3, ..., k_x)$ as interpretation anchors instead of only one basis as Harmonica does. We calculate $k_x$ interpretation models $g_{x,i}$ on each anchor and denote the coefficient of $g_{x,i}$ as $\alpha_{x,i}$. For simplicity, we omit the subscript $x$ in the following.

The number of bases $k$ is highly related to consistency, so we could define the log value $\log k$ to evaluate the inconsistency of the interpretation models. The minimum inconsistency value is $0$, as in Harmonica ($k = 1$), while for attribution methods, the inconsistency should be almost $n$, which is the number of input variables. The maximum inconsistency value depends on the neighborhood region $\mathcal{N}_x$, since the number of interpretation models should not exceed the number of points in $\mathcal{N}_x$. As a special case, the Harmonica algorithm achieves $0$ inconsistency, making it the most consistent algorithm.

However, if one only considers the number of different interpretation algorithms, users may still encounter a high level of inconsistency empirically when the interpretations significantly differ from one another. Thus, another critical constraint to add is the spectrum distance constraint among different interpretations of different anchors. Based on Theorem 4, we introduce Harmonica-anchor-constrained in Algorithm 3 (in Appendix). In this algorithm, $\lambda_2$ serves as a penalty coefficient for the difference between $g_i$, and we solve the problem using iterative gradient descent. Intuitively, a larger

$\lambda_2$ restricts the expressive power of the interpretation models $g_i$, leading to a larger interpretation error.

## 5 Experiments

### 5.1 Analysis on Polynomial Functions

To investigate the performance of different interpretation methods, we *manually* examine the output of various algorithms including LIME [47], SHAP [38], Shapley Interaction Index [45], Shapley Taylor [61, 26], Faith-SHAP [62], Harmonica, and Low-degree (Appendix D) for lower-order polynomial functions.

We observe that all algorithms can accurately learn the coefficients of the first-order polynomial. For the second-order polynomial function, only Shapley Taylor, Faith-SHAP, Harmonica, and Low-degree can learn all the coefficients accurately. For the third-order polynomial function, only Faith-SHAP, Harmonica, and Low-degree succeed. Due to space constraints, we defer the details to Appendix F.

### 5.2 Experimental Setup

In the rest of this section, we conduct experiments to evaluate the interpretation error $\mathbb{I}_{p,\mathcal{N}_x}(f, g)$ and truthful gap $\mathbb{T}_{V_C}(f, g)$ of Harmonica and other baseline algorithms on language and vision tasks quantitatively. In our experiments, we choose 2nd order and 3rd order Harmonica algorithms, which correspond to setting $C$ to include all terms with order at most 2 and 3.

The baseline algorithms chosen for comparison include LIME [47], Integrated Gradients [60], SHAP [38], Integrated Hessians [30], Shapley Taylor interaction index, and Faith-SHAP, where the first three are first-order algorithms, and the last three are second-order algorithms.

The two language tasks we select are the SST-2 [55] dataset for sentiment analysis and the IMDb [40] dataset for movie review classification. The vision task is the ImageNet [33] for image classification. To demonstrate the capability of our interpretation framework applied to vision tasks, we have generated two examples in Figure 2, compared with LIME, Integrated Gradients (IG), and SHAP. Note that all of these methods are applied to the same ground-truth image segmentation provided by the MS-COCO dataset [35]. For ablations on using the SLIC superpixels [2], please refer to Appendix H.

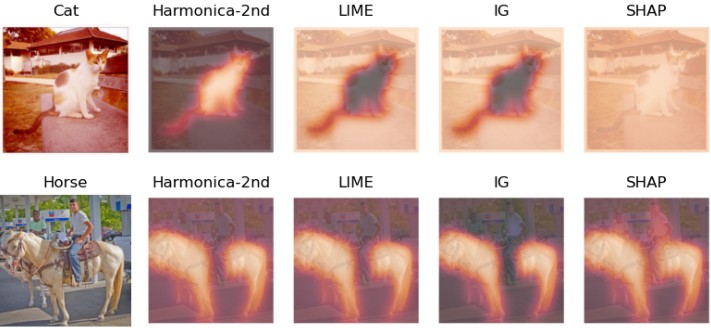

Figure 2: Illustrative examples for applying our interpretation method on MS-COCO dataset.

For the SST-2 dataset, we attempt to interpret a convolutional neural network (see details in Appendix G) trained with the Adam [32] optimizer for 10 epochs. The IMDb dataset contains long paragraphs, and each paragraph has multiple sentences. By default, we use periods, colons, and exclamations to separate sentences. For the ImageNet [33] dataset, we aim to provide class-specific interpretation, meaning that only the class with the maximum predicted probability is considered for each sample. We use the official pre-trained ResNet-101 [28] model from PyTorch.

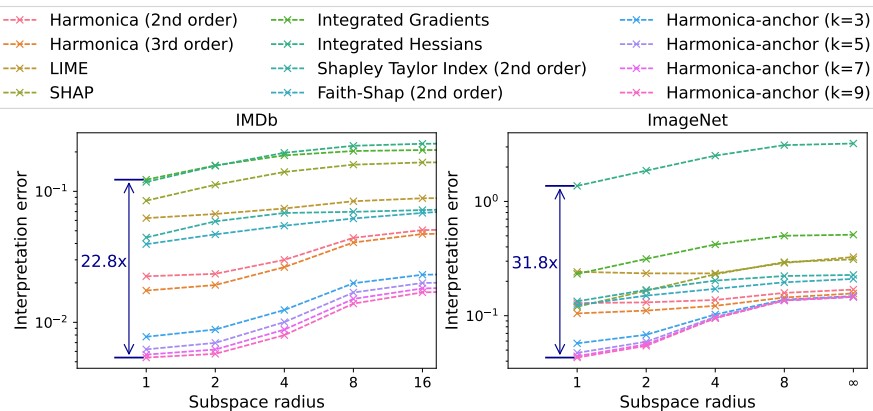

Figure 3: Visualization of $L^2$ interpretation error $\mathbb{I}_{p,\mathcal{N}_x}(f,g)$ of several state-of-the-art interpretation methods evaluated on IMDb and ImageNet datasets.

## 5.3 Results on Interpretation Error

For a given input sentence $x$ with length $l$, we define the induced neighborhood $\mathcal{N}_x$ by introducing a masking operation on this sentence. The radius $0 \leq r \leq l$ is defined as the maximum number of masked words.

Figure 3 displays the $L^2$ interpretation error evaluated under different neighborhoods with a radius ranging from 1 to $\infty$ for all the considered datasets. Here, $\infty$ represents the maximum sentence length, which may vary for different data points. We also inspect $L^1$ and $L^0$ norms. Here, $L^2$ and $L^1$ are defined according to Eqn. (2) with $p = 2$ and $p = 1$, respectively. And $L^0$ denotes $\int_{\mathcal{X}} \mathbb{1}|f(x) - g(x)| \geq 0.1 d\mu(x)$. We can see that Harmonica consistently outperforms all the other baselines on all radii.

Note that for IMDb in Figure 3, we make a slight modification such that the masking operation is performed on sentences in one input paragraph instead (we also change the definition of radii accordingly). For ImageNet in Figure 3, the interpretation error is evaluated on 1000 random images, and the masking operation is performed on 16 superpixels in one input image instead (we also change the definition of radii accordingly). We can see that when the neighborhood's radius is greater than 1, Harmonica outperforms all the other baselines. Specifically, when evaluated on ImageNet dataset, Harmonica-anchor ($k = 9$) achieves 31.8 times lower interpretation error $\mathbb{I}_{2,\mathcal{N}_1}$ compare to Integrated Gradients [60]. Limited by space, the detailed numerical results and interpretation error under other norms are presented in Appendix I. In contrast, Harmonica-local achieves more accurate local consistency but high error outside the neighborhood. The relevant results are deferred in Appendix J due to limited space.

As mentioned in Section 4, Harmonica-anchor further reduces the interpretation error by learning each subspace separately. Figure 3 shows that the $L_2$ interpretation error of Harmonica-anchor achieves lower error than Harmonica on various datasets, and the error further declines as we increase the number of anchors. Full results of other norms and the results of Harmonica-anchor-constrained are shown in Appendix L.

Table 1: Comparison of $C^3$ truthful gap $\mathbb{T}_C(f,g)$ results evaluated on IMDb and ImageNet datasets (lower truthful gap means better truthfulness of the interpretation).

| Method | Harm. 2nd | Harm. 3rd | LIME | SHAP | IG | IH | Shapley Taylor | Faith-Shap |
|---|---|---|---|---|---|---|---|---|
| IMDb | 0.540 | **0.174** | 5.343 | 1.390 | 1.438 | 1.948 | 7.005 | 15.528 |
| ImageNet | 0.660 | **0.246** | 0.738 | 2.023 | 1.848 | 175.368 | 0.474 | 0.430 |

## 5.4 Results on Truthful Gap

For convenience, we define the set of bases $C^d$ up to degree $d$ as $C^d = \{\chi_S | S \subseteq [n], |S| \leq d\}$. We evaluate the truthful gap on the set of bases $C^3$, $C^2$, and $C^1$. For implementation details on

calculating the truthful gap, please refer to Appendix K. Table 1 shows the $C^3$ truthful gap evaluated on different datasets. We can see that Harmonica outperforms all the other baseline algorithms. Due to space limitations, $C^2$ and $C^1$ results are provided in Appendix K.

## 5.5 More Experimental Results

For more experimental results on different image segmentation methods, different neural network architectures, and different choices of baselines, please refer to Appendix H. For more results on the Low-degree algorithm, please refer to Appendix M.

# 6 Related Work

Interpretability is a critical topic in machine learning, and we refer the reader to [20, 37] for insightful general discussions. Below we discuss different types of interpretable models.

**Removal-based explanations.** Covert et al. [18] presents a unified framework for removal-based model explanation methods, connecting 26 existing techniques such as LIME [47], SHAP [38], Meaningful Perturbations [21], and permutation tests [10]. LIME [47] is a classical method that samples data points following a predefined sampling distribution and computes a function that empirically satisfies local fidelity.

The Shapley value [50, 72, 24] originates from cooperative game theory and is used to allocate the total value generated by a group of players among individual players. It is the unique kind of method that satisfies a few important properties, including efficiency, symmetry, dummy, additivity, etc. Shapley values have been extensively applied to machine learning model explanations [38, 39, 57, 59, 67, 78, 23, 77] and feature importance [17]. Recent research has focused on developing efficient approximation methods for Shapley values [38, 15, 5, 16, 31, 26, 68]. Many works have generalized Shapley values to higher-order feature interactions [45, 24, 61, 41, 63, 1, 62].

**Gradient-based explanations.** Gradient-based explanations are popular for deep learning models, such as CNNs. These methods include SmoothGrad, Integrated Gradient, GradCAM, DeepLift, LRP, etc. [53, 54, 52, 60, 74, 49, 6, 51, 13, 42, 52, 48]. Although these methods have seen extensive use in various areas, gradient-based methods are often time-consuming and can be insensitive to random model parameterization [3].

**Model-specific interpretable models.** Interpretable or transparent (white-box) models are inherently ante-hoc and model-specific. One primary goal of utilizing interpretable models is to achieve inherent model interpretability. Prominent approaches include Decision Trees ([66, 7, 75]), Decision Rules ([69, 58]), Decision Sets ([34, 70]), and Linear Models ([64, 65]). Moreover, another research direction attempts to use an interpretable model surrogate to approximate the original black-box models. [14] employs a two-layer additive risk model for interpreting credit risk assessments. [8] suggests an approach called model extraction that greedily learns a decision tree to approximate $f$. However, these methods primarily rely on heuristics and lack theoretical guarantees.

# 7 Conclusion

In this paper, we tackled the problem of generating consistent interpretations and introduced the impossible trinity theorem of interpretability, under a formal framework for understanding the interplay between interpretability, consistency, and efficiency. Since a consistent interpretation cannot be efficient, we relaxed efficiency to truthfulness, meaning the interpretation matches the target function in a specific subspace. This led to the problem of learning Boolean functions and the proposal of new algorithms based on the Fourier spectrum and a localized version of Harmonica. Our methods showed lower interpretation errors and improved consistency compared to the existing approaches.

While our work offers theoretical insights, many open questions and challenges remain in building more interpretable, consistent, and efficient models. We hope our work serves as a foundation for future research in the area of explainable AI.

## Acknowledgments

Thanks to Jiaye Teng for the useful discussions. This work is supported by the Ministry of Science and Technology of the People's Republic of China, the 2030 Innovation Megaprojects "Program on New Generation Artificial Intelligence" (Grant No. 2021AAA0150000).

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

**Pseudo Code for Harmonica-anchor-constrained**

---

**Algorithm 3:** Harmonica-anchor-constrained

---

**Input:** anchor number $k$, model $f$, a distance metric function $d(\cdot, \cdot) : (\chi_S, \chi_S) \to \mathbb{R}$ which calculates distance between two anchors (e.g., Hamming distance or $L_p$ norm), sampling number $T$, loss balance coefficient $\lambda_1, \lambda_2$, update epochs $E$, update rate $\eta$

**Output:** interpretation models $g_i$ with coefficients $\alpha_i$, $i = 1, 2, 3, ..., k$.

1: Fix random bases $b_i, i = 1, 2, 3, ..., k$ as interpretation anchors.
2: Randomly initialize $\alpha_i \in \mathbb{R}^{|C|}, i = 1, 2, 3, ..., k$.
3: Randomly sample $T$ bases $b_1, b_2, ..., b_T$ and accordingly, we calculate $f(x_1), f(x_2), ..., f(x_T)$.
4: Assign each basis to the anchor with minimal distance and index with
  $d_i = \min\{\arg\min_j d(b_i, b_j^x)\}, i = 1, 2, 3, ...T$ and $j = 1, 2, 3, ..., k$.
5: **for** $e = 1, 2, 3, ..., E$ **do**
6:    $L_f = \frac{1}{T} \sum_{i=1}^{T} \left( \sum_{S, \chi_S \in C} \alpha_{d_i, S} \chi_S(x_i) - f(x_i) \right)^2$
7:    $L_r = \frac{1}{k} \sum_{i=1}^{k} \|\alpha_i\|_1$.
8:    $L_c = \frac{1}{k} \sum_{i=1}^{k} \|\alpha_i - \frac{1}{k} \sum_{j=1}^{k} \alpha_j\|_2$.
9:    Update $\alpha_i$ with $\alpha_i \leftarrow \alpha_i - \eta \frac{\partial(L_f + \lambda_1 L_r + \lambda_2 L_c)}{\partial \alpha_i}$.
10: **end for**

---

# A   Fourier Analysis of Boolean Function

Fourier analysis of Boolean function is a fascinating field, and we refer the reader to [43] for a more comprehensive introduction. We define the inner product as follows.

**Definition A.1** (Inner product). Given two functions $f, g \in \{-1, 1\}^n \to \mathbb{R}$, their inner product is:

$$\langle f, g \rangle \triangleq \mathbb{E}_{x \sim \{-1,1\}^n}[f(x)g(x)] = 2^{-n} \sum_{x \in \{-1,1\}^n} f(x)g(x). \tag{9}$$

In addition, we define the induced norm $\|f\|_2 \triangleq \sqrt{\langle f, f \rangle}$, and more generally, for $p > 0$,

$$\|f\|_p \triangleq \mathbb{E}[|f(x)|^p]^{1/p}. \tag{10}$$

The inner product defines one kind of similarity between two functions and is invariant under different basis. Specifically, we have the following Theorem.

**Definition A.2** (Plancherel's Theorem). Given two functions $f, g \in \{-1, 1\}^n \to \mathbb{R}$,

$$\langle f, g \rangle = \sum_{S \subseteq [n]} \hat{f}_S \hat{g}_S.$$

When setting $f = g$, we get the Parseval's identity: $\mathbb{E}[f^2] = \sum_S \hat{f}_S^2$.

A distribution over a discrete domain $S$ is often represented as a non-negative function $f : S \to \mathbb{R}^+$ which is normalized in $L_1$, i.e., $\sum_{x \in S} f(x) = 1$.

**Definition A.3** (Fourier $p$-norm). For any Boolean function $f \in \{-1, 1\}^n \to \mathbb{R}$, we define the Fourier (or spectral) $p$-norm of $f$ as

$$\hat{\|}f\hat{\|}_p \triangleq \left( \sum_{S \subseteq [n]} |\hat{f}_S|^p \right)^{1/p}. \tag{11}$$

**Definition A.4** (Fourier $p$-distance). For any Boolean function $f, g \in \{-1, 1\}^n \to \mathbb{R}$, we define the Fourier (or spectral) $p$-distance between $f$ and $g$ as

$$\hat{\|}f - g\hat{\|}_p \triangleq \left( \sum_{S \subseteq [n]} |\hat{f}_S - \hat{g}_S|^p \right)^{1/p}. \tag{12}$$

Using Fourier $p$-norm, we could rephrase Parseval's identity as: $\|f\|_2 = \hat{\|}f\hat{\|}_2$.

## B  Proofs

**Proof of Theorem 1.**

*Proof.* Pick $f \in \mathcal{F} \setminus \mathcal{G}$. If $\mathcal{A}$ is consistent with respect to $f$, let $g = \mathcal{A}(f, x) \in \mathcal{G}$ for any $x \in \mathcal{X}$. If for every $x \in \mathcal{X}$, $g(x) = f(x)$, we know $g = f \notin \mathcal{G}$, this is a contradiction. Therefore, there exists $x \in \mathcal{X}$ such that $g(x) \neq f(x)$. $\qquad\square$

**Proof of Lemma 3.4.**

*Proof.* Denote the complement space as $V_{\bar{C}}$. We may expand $f, g, v$ on both bases and get:

$$\|f - g\|^2 - \inf_{v \in V} \|f - v\|_2 = \sum_{S \in C} \langle f - g, \chi_S \rangle^2 +$$

$$\sum_{S \in \bar{C}} \langle f, \chi_S \rangle^2 - \inf_{v \in V_C} \left( \sum_{S \in C} \langle f - v, \chi_S \rangle^2 + \sum_{S \in \bar{C}} \langle f, \chi_S \rangle^2 \right)$$

$$= \sum_{S \in C} \langle f - g, \chi_S \rangle^2 - \inf_{v \in V_C} \left( \sum_{S \in C} \langle f - v, \chi_S \rangle^2 \right)$$

$$= \sum_{S \in C} \langle f - g, \chi_S \rangle^2,$$

where the last equality holds because we can set the Fourier coefficients $\hat{v}_S = \hat{f}_S$ for every $S \in C$, which further gives $\langle f - v, \chi_S \rangle = 0$. $\qquad\square$

**Proof of Theorem 3.**

*Proof.* Taking $\mu$ as the uniform distribution, then $\mathbb{I}_0(f, g) = \frac{\sum_{x \in \{-1,1\}^n} \mathbf{1}_{f(x) \neq g(x)}}{2^n} = \frac{|\mathrm{supp}(f-g)|}{2^n}$. Note $\mathbb{D}_0(f, g) = \sum_{S \subseteq [n]} \mathbf{1}_{\hat{f}(S) \neq \hat{g}(S)} = |\mathrm{supp}(\hat{f} - \hat{g})|$. The conclusion follows from Proposition C.1. $\qquad\square$

**Proof of Proposition 3.6.**

*Proof.* From the definition of $\mathbb{I}_2(f, g)$ and $\mathbb{D}_2(f, g)$. By Parseval's identity, $\mathbb{I}_2^2(f, g) = 2^{-n} \sum_{x \in \{-1,1\}^n} (f(x) - g(x))^2 = \sum_{S \subset [n]} (\hat{f}(S) - \hat{g}(S))^2 = \mathbb{D}_2^2(f, g)$. $\qquad\square$

**Proof of Theorem 4.**

*Proof.* From Proposition 3.6 (Parseval's identity), we have (notice that our functions $g_i$ are defined on $\mathcal{X} = \{-1, 1\}^n$ globally and then restricted to $\mathcal{X}_i \subseteq \mathcal{X}$).

$$\hat{\|}g_i|_{x \in \mathcal{X}_i} - \bar{g}|_{x \in \mathcal{X}_i}\hat{\|}_2 = \hat{\|}g_i|_{x \in \mathcal{X}} - \bar{g}|_{x \in \mathcal{X}}\hat{\|}_2 = \|g_i|_{x \in \mathcal{X}} - \bar{g}|_{x \in \mathcal{X}}\|_2 \geq \|g_i|_{x \in \mathcal{X}_i} - \bar{g}|_{x \in \mathcal{X}_i}\|_2.$$

Let us employ the triangle inequality for norms. Specifically, for each $i = 1, 2, \ldots, N$, we apply the triangle inequality as follows:

$$\|f|_{x \in \mathcal{X}_i} - g_i|_{x \in \mathcal{X}_i}\|_2 + \|g_i|_{x \in \mathcal{X}_i} - \bar{g}|_{x \in \mathcal{X}_i}\|_2 \geq \|f|_{x \in \mathcal{X}_i} - \bar{g}|_{x \in \mathcal{X}_i}\|_2.$$

Summing this inequality over all $i$ yields:

$$\sum_{i=1}^N \|f|_{x \in \mathcal{X}_i} - g_i|_{x \in \mathcal{X}_i}\| + \sum_{i=1}^N \|g_i|_{x \in \mathcal{X}_i} - \bar{g}|_{x \in \mathcal{X}_i}\| \geq \sum_{i=1}^N \|f|_{x \in \mathcal{X}_i} - \bar{g}|_{x \in \mathcal{X}_i}\|_2.$$

By the definitions of $\mathbb{I}_{\text{total}}$ and $\mathbb{D}_{\text{total}}$, we arrive at:

$$\mathbb{I}_{\text{total}} + \mathbb{D}_{\text{total}} \geq \sum_{i=1}^{N} \| f|_{x \in \mathcal{X}_i} - \bar{g}|_{x \in \mathcal{X}_i} \|_2.$$

This completes the proof. $\qquad\square$

### B.1 Harmonica Algorithm

To present the theoretical guarantees of the Harmonica algorithm, we introduce the following definition, which is slightly different from its original version in [27].

**Definition B.1** (Approximately sparse function). We say a function $f \in \{-1,1\}^n \to \mathbb{R}$ is $(\epsilon, s, C)$-bounded, if $\mathbb{E}[(f - \sum_{\chi_S \in C} \hat{f}(S)\chi_S)^2] \leq \epsilon$ and $\sum_S |\hat{f}(S)| \leq s$.

Here $f$ is $(\epsilon, s, C)$-bounded means it is almost readable and has bounded $\ell_1$ norm. Our algorithm is slightly different from the original algorithm proposed by [27], but similar theoretical guarantees still hold, as stated below.

**Proof of Theorem 2.** Our proof is similar to the one in the original paper [27], with changes in the readable notion, which is now more flexible than being low order. First, recall the classical Chebyshev inequality.

**Theorem B.2** (Multidimensional Chebyshev inequality). *Let $X$ be an $m$ dimensional random vector, with expected value $\mu = \mathbb{E}[X]$, and covariance matrix $V = \mathbb{E}\left[(X - \mu)(X - \mu)^T\right]$. If $V$ is a positive definite matrix, for any real number $\delta > 0$ :*

$$\mathbb{P}\left( \sqrt{(X - \mu)^T V^{-1} (X - \mu)} > \delta \right) \leq \frac{m}{\delta^2}.$$

*Proof of Theorem 2.* Let $f$ be an $(\varepsilon/4, s, C)$-bounded function written in the orthonormal basis as $\sum_S \hat{f}(S)\chi_S$. We can equivalently write $f$ as $f = h + g$, where $h$ is supported on $C$ that only includes coefficients of magnitude at least $\varepsilon/4s$ and the constant term of the polynomial expansion of $f$.

Since $L_1(f) = \sum_S \left| \hat{f}_S \right| \leq s$, we know $h$ is $4s^2/\varepsilon + 1$ sparse. The function $g$ is thus the sum of the remaining $\hat{f}(S)\chi_S$ terms not included in $h$. Denote the set of bases that appear in $C$ but not in $g$ as $R$, so we know the coefficient of $f$ on the bases in $R$ is at most $\epsilon/4s$.

Draw $m$ (to be chosen later) random labeled examples $\left\{ \left( z^1, y^1 \right), \ldots, \left( z^m, y^m \right) \right\}$ and enumerate all $N = |C|$ basis functions $\chi_S \in C$ as $\{\chi_1, \ldots, \chi_N\}$. Form matrix $A$ such that $A_{ij} = \chi_j \left( z^i \right)$ and consider the problem of recovering $4s^2/\varepsilon + 1$ sparse $x$ given $Ax + e = y$ where $x$ is the vector of coefficients of $h$, the $i$ th entry of $y$ equals $y^i$, and $e_i = g\left(z^i\right)$.

We will prove that with constant probability over the choice $m$ random examples, $\|e\|_2 \leq \sqrt{\varepsilon m}$. Applying Theorem 5 in [27] by setting $\eta = \sqrt{\varepsilon}$ and observing that $\sigma_{4s^2/\varepsilon+1}(x)_1 = 0$ (see definition in the theorem), we will recover $x'$ such that $\|x - x'\|_2^2 \leq c_2^2 \varepsilon$ for some constant $c_2$. As such, for the function $\tilde{f} = \sum_{i=1}^N x_i' \chi_i$ we will have $\mathbb{E}\left[ \|h - \tilde{f}\|^2 \right] \leq c_2^2 \varepsilon$ by Parseval's identity. Note, however, that we may rescale $\varepsilon$ by a constant factor $1/\left(2c_2^2\right)$ to obtain error $\varepsilon/2$ and only incur an additional constant (multiplicative) factor in the sample complexity bound. By the definition of $g$, we have

$$\|g\|^2 = \left( \sum_{S, \chi_S \notin C} \hat{f}(S)^2 + \sum_{S \in R} \hat{f}(S)^2 \right). \tag{13}$$

where each $\hat{f}(S)$ for $S \in R$ is of magnitude at most $\varepsilon/4s$. By Fact 4 in [27] and Parseval's identity we have $\sum_R \hat{f}(R)^2 \leq \varepsilon/4$. Since $f$ is $(\varepsilon/4, s, C)$-concentrated we have $\sum_{S, \chi_S \notin C} \hat{f}(S)^2 \leq \varepsilon/4$. Thus, $\|g\|^2$ is at most $\varepsilon/2$. Therefore, by triangle inequality $\mathbb{E}\left[ \|f - \tilde{f}\|^2 \right] \leq \mathbb{E}\left[ \|h - \tilde{f}\|^2 \right] + \mathbb{E}\left[ \|g\|^2 \right] \leq \varepsilon$. It remains to bound $\|e\|_2$. Note that since the examples are chosen independently, the entries

$e_i = g\left(z^i\right)$ are independent random variables. Since $g$ is a linear combination of orthonormal monomials (not including the constant term), we have $\mathbb{E}_{z \sim D}[g(z)] = 0$. Here we can apply linearity of variance (the covariance of $\chi_i$ and $\chi_j$ is zero for all $i \neq j$ ) and calculate the variance

$$\mathrm{Var}\left(g\left(z^i\right)\right) = \left(\sum_{S, \chi_S \notin C} \hat{f}(S)^2 + \sum_{S \in R} \hat{f}(S)^2\right).$$

With the same calculation as Eqn. (13), we know $\mathrm{Var}\left(g\left(z^i\right)\right)$ is at most $\varepsilon/2$. Now consider the covariance matrix $V$ of the vector $e$ which equals $\mathbb{E}\left[ee^\top\right]$ (recall every entry of $e$ has mean 0). Then $V$ is a diagonal matrix (covariance between two independent samples is zero), and every diagonal entry is at most $\varepsilon/2$. Applying Theorem B.2 we have

$$\mathbb{P}\left(\|e\|_2 > \sqrt{\frac{\varepsilon}{2}}\delta\right) \leq \frac{m}{\delta^2}.$$

Setting $\delta = \sqrt{2m}$, we conclude that $\mathbb{P}\left(\|e\|_2 > \sqrt{\varepsilon m}\right) \leq \frac{1}{2}$. Hence with probability at least $1/2$, we have that $\|e\|_2 \leq \sqrt{\varepsilon m}$. From Theorem 5 in [27], we may choose $m = \tilde{O}\left(s^2/\varepsilon \cdot \log n^d\right)$. This completes the proof. Note that the probability $1/2$ above can be boosted to any constant probability with a constant factor loss in sample complexity.

For the running time complexity, we refer to [4] for optimizing linear regression with $\ell_1$ regularization. The running time is $O((T \log \frac{1}{\epsilon} + L/\epsilon) \cdot |C|)$, where $L$ is the smoothness of each summand in the objective. Since each $\chi_S$ takes value in $\{-1, 1\}$, the smoothness is bounded by the number of entries in each summand, which is $|C|$. Therefore, the running time is bounded by $O((T \log \frac{1}{\epsilon} + |C|/\epsilon) \cdot |C|)$. $\quad\square$

## C  Uncertainty Principle for Boolean functions

In the field of modern physics, the state of a particle on a given domain $S$ is represented by a complex function on that domain, and the probability of finding the particle in a specific position $x$ on $S$ is given by the square of the modulus of the function evaluated at $x$. To ensure that the function is normalized, in the case where $S$ is continuous, the function must satisfy $\int_{x \in \mathbb{R}} |f(x)|^2 dx = 1$. The Fourier transform of the function, denoted by $\hat{f}$, is also normalized in $L_2$ under a unitary transformation, and $|\hat{f}(x)|^2$ represents the probability density function of the distribution of the particle's momentum.

The Heisenberg uncertainty principle states that the product of the variances of a particle's position and momentum is at least one, with an appropriate choice of units. This principle has physical significance and relates a function on $\mathbb{R}$ to its Fourier transform.

In 1957, [29] proposed an entropic form of the uncertainty principle, which was later proven nearly two decades later by Beckner [9]. The inequality states that $H_e[f] + H_e[\hat{f}] \geq 1 - \ln 2$, where $H_e[f]$ is the differential entropy of $f$, defined as $-\int_{x \in \mathbb{R}} |f(x)|^2 \ln |f(x)|^2 dx$.

When the domain is $\mathbb{F}_2^n$ (equivalently, $\{-1, 1\}^n$), a similar inequality [25] holds with a different constant. Let $f : \mathbb{F}_2^n \to \mathbb{C}$ have a Fourier transform $\hat{f} : \mathbb{F}_2^n \to \mathbb{C}$. Then,

$$H\left[\frac{f}{\|f\|}\right] + H\left[\frac{\hat{f}}{\|\hat{f}\|}\right] \geq n,$$

where $H[f] = -\sum_{x \in \mathbb{F}_2^n} |f(x)|^2 \log_2 |f(x)|^2$, and $\|f\| = \sqrt{\sum_{x \in \mathbb{F}_2^n} f(x)^2}$.

Now we present the discrete uncertainty principle for $\mathbb{Z}_2^n$ as the following. It can be proved using Theorem 23 in Dembo, Cover, and Thomas [19].

All the Fourier coefficients together are called the Fourier spectrum of $f$. We can define the following distance between two functions based on their spectrums.

As there may be many candidates $p$ for evaluating interpretation error and spectrum distance. In the following, we will investigate the most natural choice for $p$. As the Fourier expansion uniquely determined a Boolean function, we would like the distance between their expansion to have the same tendency as the interpretation error, thus the spectrum can encode enough information to reflect the accuracy of the interpretation. We shall first discuss the most "strict" case, i.e. the $L_0$ norm.

We shall introduce some notations for better understanding. Denote the support of an Boolean function $f$ as $\mathrm{supp}(f) \triangleq \{x \in \{-1, 1\}^n \mid f(x) \neq 0\}$. The support of Fourier coefficients (Fourier spectrum), is defined by $\mathrm{supp}(\hat{f}) \triangleq \{S \subseteq [n] \mid \hat{f}(S) \neq 0\}$. The uncertainty principle for Boolean functions is stated as follows.

**Proposition C.1** (Uncertainty Principle for Boolean functions [25, 43])**.** *For every nonzero function* $f : \{-1, 1\}^n \to \mathbb{R}$,

$$|\mathrm{supp}(f)| \cdot |\mathrm{supp}(\hat{f})| \geq 2^n.$$

# D  Low-degree Algorithm

The low-degree algorithm is based on the concentration inequality, and it estimates the coefficient of each axis individually.

---
**Algorithm 4:** Low-degree
---
1. Given uniformly randomly sampled $x_1, \cdots, x_T$, evaluate them on $f$: $\{f(x_1), ...., f(x_T)\}$.
2. For any $\chi_S \in C$, let $\hat{g}_S = \frac{\sum_{i=1}^T f(x_i)\chi_S(x_i)}{T}$.
3. Output the polynomial $g(x) = \sum_{S, \chi_S \in C} \hat{g}_S \chi_S(x)$.

---

**Theorem D.1** ([36])**.** *Given any* $\epsilon, \delta > 0$, *assuming that function* $f$ *is bounded by* $B$, *when* $T \geq \frac{2B^2}{\epsilon^2} \log \frac{2|C|}{\delta}$, *we have*

$$\Pr\left[\forall \chi_S \in C, s.t., |\hat{g}_S - \hat{f}_S| \leq \epsilon\right] \geq 1 - \delta.$$

Theorem D.1 was proved using the Hoeffding bound, and we included the proof here for completeness.

*Proof.* Since we are given $T$ samples to estimate $\hat{f}(S)$ for every $S$, we can directly apply the Hoeffding bound (notice that the function is bounded by $B$):

$$\Pr\left(|\alpha_S - \hat{f}(S)| \geq \epsilon\right) \cdot |C| \leq 2\exp\left(-\frac{2T\epsilon^2}{4B^2}\right) = 2\exp\left(-\frac{T\epsilon^2}{2B^2}\right).$$

Notice that $T \geq \frac{2B^2}{\epsilon^2} \log \frac{2|C|}{\delta}$, we know the right hand size is bounded by $\frac{\delta}{|C|}$, so Theorem D.1 is proved. $\square$

**Remarks**. The theoretical guarantee of Harmonica assumes the target function $f$ is approximately sparse in the Fourier space, which means most of the energy of the function (Fourier coefficients) is concentrated in the bases in $C$. This is not a strong assumption, because if $f$ is not approximately sparse, it means $f$ has energy in many different bases, or more specifically, the bases with higher orders. In other words, $f$ has a large variance and is difficult to interpret. In this case, no existing algorithms will be able to give consistent and meaningful interpretations.

Likewise, although the Low-degree algorithm does not assume sparsity for $f$, it cannot learn all possible functions accurately as well. There are $2^n$ different bases, and if we want to learn the coefficients for all of them, the cumulative error for $g$ is at the order of $\Omega(2^n \epsilon)$, which is exponentially large. This is not surprising due to the no free lunch theorem in the generalization theory, as we do not expect to be able to learn "any functions" without exponentially many samples.

# E  Discussion on the Existing Algorithms

In this section, we compare our approaches with the existing techniques from different perspectives of interpretation error (efficiency), truthfulness, and consistency.

**LIME [47].**  Given an input $x$, Lime samples the neighborhood points based on a sampling distribution $\Pi_x$, and optimizes the following program:

$$\min_{g \in \mathcal{G}} L(f, g, \Pi_x) + \Omega(g).$$

where $L$ is the loss function describing the distance between $f$ and $g$ on the sampled data points, $\mathcal{G}$ is the set of readable functions (e.g. the set of linear functions), $\Omega(\cdot)$ is a function that characterizes the complexity of $g$. In other words, LIME tries to minimize the fitting error and simultaneously minimizes the complexity of $g$ (which is usually the sparsity of the linear function). By minimizing $L$, LIME also works towards minimizing the interpretation error, but their approach is purely heuristic, without any theoretical guarantees. Although their readable function set can easily generalize to the set with higher order terms, the sampling distribution $\Pi_x$ is not uniform, so it is difficult to incorporate the orthonormal basis into their framework. In other words, the model they compute is not truthful.

**Attribution methods.**  As we discussed in the introduction, attribution methods mainly focus on individual inputs, instead of the neighboring points. Therefore, it is difficult for the attribution methods to achieve low inconsistency, especially for first-order methods like SHAP [38] and IG [60].

Consider SHAP [38] as a motivating example, illustrated in Figure 1 for the task of sentiment analysis of movie reviews. In this example, the interpretations of two slightly different sentences are inconsistent. This inconsistency arises not only because the weights of each word differ significantly, but also because after removing the word "very" with a weight of $15.0\%$, the network's output only drops by $6.6\%$. In other words, **the interpretation does not explain the network's behavior even in a small neighborhood of the input**.

For higher-order attribution methods, consistency can potentially be improved due to their enhanced representation power. The classical Shapley interaction index has the problem of not precisely fitting the underlying function, as observed by [61], who proposed Shapley Taylor interaction index [61] with better empirical performance. Shapley Taylor interaction index satisfies the generalized efficiency axiom, which says for all $f \in \{-1, 1\}^n \to \mathbb{R}$,

$$\sum_{S \subseteq [n], |S| \leq k} \mathcal{I}_S^k(f) = f([n]) - f(\emptyset).$$

We should remark that both the Shapley interaction index and Shapley Taylor interaction index were not originally designed for consistent interpretations, so they did not specify how to generalize the interpretation for the neighboring points. To this end, we make a global extension to the Shapley value based interpretation, that is, using Shapley interaction indices or Shapley Taylor interaction indices as the coefficients of corresponding terms of the polynomial surrogate function.

$$g(x_1, x_2, \cdots, x_n) = f(\emptyset) + \sum_{x_i \in S, S \subseteq [n]} \mathcal{I}(f, S).$$

However, these higher-order Shapley value based methods all focus on the original Shapley value framework, so their interpretations are not truthful, i.e., not getting the exact coefficient of $f$ even on the "simple bases". Moreover, as we will show in our experiments, higher-order methods still incur high interpretation errors compared with our methods.

When applying Shapley value techniques for visual search, [26] proposed an interesting and novel sampling $+$ Lasso regression algorithm for efficiently computing higher order Shapley Taylor index in their experiments. However, their methods are based on a sampling probability distribution generated from permutation numbers, which is far from the uniform distribution. Additionally, their algorithm is based on the Shapley Taylor index, so their method is not truthful as well.

**Discussion on universal consistency.**  When discussing consistency, there are two distinct settings: "global consistency" and "universal consistency." This paper mainly focuses on global consistency (Definition 2.2), where the interpretation pertains only to input features and not to others. This scenario falls under the category of "removal-based explanations," and most existing interpretation methods belong to this category (26 of them were discussed in [18]). In contrast, universal consistency implies that the interpretation may depend on features different from the input features. An example of interpreting the sentence "I am happy" could be, "This sentence does not include [very], so this person is not extremely happy." Universal consistency is more challenging than global consistency, and we hypothesize that more powerful machinery, such as foundation models, is needed.

**Emergence of self-explainability in foundation models.** In the groundbreaking work Sparks of AGI [11], the authors conducted comprehensive experiments on GPT-4's self-explainability. Chain-of-thought reasoning [73, 71, 76, 79] can be considered an interpretation provided by the model itself, although current LLMs are not always to be consistent or truthful. In the long run, however, this approach may represent the ultimate path to explainable AI. In the computer vision area, DINO [12, 44] emerges the ability of self-explanation provided by the attention map inside the model, which can be directly utilized as heatmap visualization or even applied to segmentation tasks.

**Proposition E.1** (Omniscient LLM is the way). *An (infinitely) large (multi-modal) language model pre-trained using (infinitely) large corpus, with zero pre-training loss and generalization loss, which has been perfectly aligned to be truthful, is interpretable, efficient, and consistent.*

*Proof.* To prove the proposition, let us examine each attribute (interpretable, efficient, and consistent) in the context of an (infinitely) large (multi-modal) language model (LLM):

1. **Interpretable**: By assumption, the LLM has zero pre-training loss and generalization loss, implying perfect knowledge representation. Furthermore, it is perfectly aligned to be truthful, satisfying the criteria for interpretability as per Definition 2.4.
2. **Efficient**: The efficiency condition $g(x) = f(x)$ is trivially satisfied because $f = g$ by the condition of self-explainability. Thus, the LLM is efficient as per Definition 2.3.
3. **Consistent**: Given that the LLM's responses are generated based on a perfect understanding and alignment, it will generate the same function $\mathcal{A}(f, x)$ for every $x \in \mathcal{X}$, thereby being consistent as per Definition 2.2.

It's crucial to note that self-explainability does not violate the Impossible Trinity (Theorem 1). When $f = g$, the LLM serves as its own interpreter, inherently satisfying all three criteria. Alternatively, if $f \subseteq g$, where $g$ has an even larger concept class than $f$, the interpretability criteria are also met, albeit this is not a practical scenario for standard interpretation algorithms.

Therefore, an (infinitely) large LLM, pre-trained with an (infinitely) large corpus and aligned perfectly to be truthful, satisfies all the conditions to be interpretable, efficient, and consistent. $\square$

# F   Test with Low order Polynomial Functions

## F.1   First order polynomial function

To investigate the performance of different interpretation methods, let us take a closer look at a 1st order polynomial function:

$$f_1(x_1, x_2, x_3) = \frac{1}{2}x_1 - \frac{1}{3}x_2 + \frac{1}{4}x_3,$$

For this simple function, we can manually compute the outcome of each algorithm, as illustrated in Table 2. If the algorithm's output is correct, i.e., equal to the output of $f_1$, we write a check mark. Otherwise, we write down the actual output of the given interpretation algorithm.

As we can see, all methods are consistent and efficient for all cases. In fact, all variants of Shapley indices degraded to 1st order Shapley values.

| Algorithms | $(-1,-1,-1)$ | $(-1,-1,+1)$ | $(-1,+1,-1)$ | $(-1,+1,+1)$ | $(+1,-1,-1)$ | $(+1,-1,+1)$ | $(+1,+1,-1)$ | $(+1,+1,+1)$ |
|---|---|---|---|---|---|---|---|---|
| Ground Truth | $-0.417$ | $+0.083$ | $-1.083$ | $-0.583$ | $+0.583$ | $+1.083$ | $-0.083$ | $+0.417$ |
| LIME | ✔ | ✔ | ✔ | ✔ | ✔ | ✔ | ✔ | ✔ |
| SHAP | ✔ | ✔ | ✔ | ✔ | ✔ | ✔ | ✔ | ✔ |
| Shapley Interaction Index (1st order) | ✔ | ✔ | ✔ | ✔ | ✔ | ✔ | ✔ | ✔ |
| Shapley Taylor Index (1st order) | ✔ | ✔ | ✔ | ✔ | ✔ | ✔ | ✔ | ✔ |
| Faith-Shap (1st order) | ✔ | ✔ | ✔ | ✔ | ✔ | ✔ | ✔ | ✔ |
| Low-degree (1st order) | ✔ | ✔ | ✔ | ✔ | ✔ | ✔ | ✔ | ✔ |
| Harmonica (1st order) | ✔ | ✔ | ✔ | ✔ | ✔ | ✔ | ✔ | ✔ |

Table 2: Interpretations by LIME, SHAP, Shapley Interaction Index, Shapley Taylor Index, Faith-Shap, Low-degree, and Harmonica on the 1st order polynomial function $f_1$.

## F.2 Second order polynomial function

In addition, let us take a closer look at a 2nd order polynomial function:

$$f_2\left(x_1, x_2, x_3\right) = \frac{1}{2}x_1 - \frac{1}{3}x_2 + \frac{1}{4}x_3 - \frac{1}{5}x_1x_2 + \frac{1}{6}x_1x_3 - \frac{1}{7}x_2x_3,$$

For this simple function, we can manually compute the outcome of each algorithm, as illustrated in Table 3. If the algorithm's output is correct, i.e., equal to the output of $f_2$, we write a check mark. Otherwise, we write down the actual output of the given interpretation algorithm.

As we can see, 2nd order interpretation algorithms, including Shapley Taylor index, Faithful Shapley, Low-degree, and Harmonica are consistent and efficient for all cases. Other methods can only fit a few inputs, the 2nd order Shapley interaction index misses all the cases because it is not efficient, and LIME misses all the cases because $f_2$ is not a linear function.

| Algorithms | $(-1,-1,-1)$ | $(-1,-1,+1)$ | $(-1,+1,-1)$ | $(-1,+1,+1)$ | $(+1,-1,-1)$ | $(+1,-1,+1)$ | $(+1,+1,-1)$ | $(+1,+1,+1)$ |
|---|---|---|---|---|---|---|---|---|
| **Ground Truth** | $-0.593$ | $-0.140$ | $-0.574$ | $-0.693$ | $+0.474$ | $+1.593$ | $-0.307$ | $+0.240$ |
| LIME | $-0.417$ | $+0.083$ | $-1.083$ | $-0.583$ | $+0.583$ | $+1.083$ | $-0.083$ | $+0.417$ |
| SHAP | $-0.240$ | $+0.283$ | $-1.250$ | $-0.726$ | $+0.726$ | $+1.250$ | $-0.283$ | ✔ |
| Shapley Interaction Index (2nd order) | $-0.329$ | $+0.171$ | $-0.995$ | $-0.781$ | $+0.671$ | $+1.505$ | $-0.395$ | $+0.152$ |
| **Shapley Taylor Index (2nd order)** | ✔ | ✔ | ✔ | ✔ | ✔ | ✔ | ✔ | ✔ |
| **Faith-Shap (2nd order)** | ✔ | ✔ | ✔ | ✔ | ✔ | ✔ | ✔ | ✔ |
| **Low-degree (2nd order)** | ✔ | ✔ | ✔ | ✔ | ✔ | ✔ | ✔ | ✔ |
| **Harmonica (2nd order)** | ✔ | ✔ | ✔ | ✔ | ✔ | ✔ | ✔ | ✔ |

Table 3: Interpretations by LIME, SHAP, Shapley Interaction Index, Shapley Taylor Index, Faith-Shap, Low-degree, and Harmonica on the 2nd order polynomial function $f_2$.

## F.3 Third order polynomial function

Finally, we investigate the following 3rd order polynomial and present the result in Table 4.

$$f_3\left(x_1, x_2, x_3\right) := \frac{1}{2}x_1 - \frac{1}{3}x_2 + \frac{1}{4}x_3 - \frac{1}{5}x_1x_2 + \frac{1}{6}x_1x_3 - \frac{1}{7}x_2x_3 + \frac{1}{8}x_1x_2x_3,$$

As we can see, Faith-Shap, Low-degree, and Harmonica are consistent and efficient for all cases. Other methods can only fit a few inputs, the 3rd order Shapley interaction index misses all the cases because it is not efficient, and LIME misses all the cases because $f_3$ is not a linear function.

| Algorithms | $(-1,-1,-1)$ | $(-1,-1,+1)$ | $(-1,+1,-1)$ | $(-1,+1,+1)$ | $(+1,-1,-1)$ | $(+1,-1,+1)$ | $(+1,+1,-1)$ | $(+1,+1,+1)$ |
|---|---|---|---|---|---|---|---|---|
| **Ground Truth** | $-0.718$ | $-0.015$ | $+0.449$ | $-0.818$ | $+0.599$ | $+1.468$ | $-0.432$ | $+0.365$ |
| LIME | $-0.417$ | $+0.083$ | $-1.083$ | $-0.583$ | $+0.583$ | $+1.083$ | $-0.083$ | $+0.417$ |
| SHAP | $-0.365$ | $+0.242$ | $-1.292$ | $-0.685$ | $+0.685$ | $+1.292$ | $-0.242$ | ✔ |
| Shapley Interaction Index (3rd order) | $-0.485$ | $+0.224$ | $-0.943$ | $-0.896$ | $+0.724$ | $+1.390$ | $-0.510$ | $+0.496$ |
| Shapley Taylor Index (3rd order) | ✔ | $+0.194$ | $-0.606$ | $-0.970$ | $+0.751$ | $+1.625$ | $-0.642$ | ✔ |
| **Faith-Shap (3rd order)** | ✔ | ✔ | ✔ | ✔ | ✔ | ✔ | ✔ | ✔ |
| **Low-degree (3rd order)** | ✔ | ✔ | ✔ | ✔ | ✔ | ✔ | ✔ | ✔ |
| **Harmonica (3rd order)** | ✔ | ✔ | ✔ | ✔ | ✔ | ✔ | ✔ | ✔ |

Table 4: Interpretations by LIME, SHAP, Shapley Interaction Index, Shapley Taylor Index, Faith-Shap, Low-degree, and Harmonica on the 3rd order polynomial function $f_3$.

Faith-SHAP has an intricate representation theorem, assigning coefficients to different terms under the Möbius transform. Since the basis induced by the Möbius transform is not orthonormal, it is unclear to us whether Faith-SHAP can theoretically compute the accurate coefficients for higher-order functions. However, the running time of Faith-SHAP has an exponential dependency on $n$, so empirically weighted sampling on subsets of features is needed [62]. This might be the main reason why our algorithms outperform Faith-SHAP in experiments on real-world datasets.

# G More on Experiment Details

In this section, we will first describe the detailed experimental settings. For the two language tasks, i.e., SST-2 and IMDb, we use the same CNN neural network. The model has a test accuracy of $85.6\%$. For the readability of results, we treat sentences as units instead of words – masking several

words in a sentence may render the entire paragraph difficult to understand and even meaningless while masking a critical sentence has meaningful semantic effects. Therefore, the radius is defined as the maximum number of masked sentences for IMDb. The word embedding layer is pre-trained by GloVe [46] and the maximum word number is set to 25, 000. Besides the embedding layer, the network consists of several convolutional kernels with different kernel sizes (3, 4, and 5). After that, we use several fully connected layers, non-linear layers, and pooling layers to process the features. A Sigmoid function is attached to the tail of the network to ensure that the output can be seen as a probability distribution. Our networks are trained with an Adam [32] optimizer with a learning rate of 1e-2 for 5 epochs. For the vision task, we choose the official ResNet [28] architecture, which is available on PyTorch and we do not discuss the architecture details here. All the experiments are run on a server with 4 Nvidia 2080 Ti GPUs. A running time comparison can be found in Table 5. More information about the run-time Python environment and implementation details can be found in our code.

Table 5: Running time of different algorithms on SST2 dataset. The main characteristics affecting the running speed are listed in the table.

| Algorithm | Main Characteristics Affecting Speed | Running Time (s) |
|---|---|---|
| Harmonica-2nd | Parallel perturbation, sample=2000 | 225 |
| Harmonica-3rd | Parallel perturbation, sample=2000 | 616 |
| LIME | Captum.attr.LimeBase, Seq. perturbation, Sample=2000, Exp. cosine sim. | 1499 |
| Integrated Gradients | Gradient step=500 | 82 |
| SHAP | Captum.attr.KernelShap, Seq. perturbation, Sample=2000 | 1235 |
| Integrated Hessians | Gradient step=50 | 3550 |
| Shapley Taylor Index | Parallel random perturbations | 651 |
| Faith-Shap | Parallel random perturbations | 538 |

The accurate representation of the high non-linearity of deep neural networks (DNNs) can be quite complex, often leading to computationally expensive or practically infeasible interpretability methods. To strike a balance between computational efficiency and faithful representation, we employ higher-order polynomial approximations to capture the non-linear aspects of the target DNNs. Specifically, we leverage Fourier basis (or polynomial basis) to represent Boolean functions over feature subsets, thus providing a compact and computationally efficient way to approximate DNNs. This is particularly useful for removal-based explanations, which inherently deal with Boolean functions representing the presence or absence of features.

It is crucial to point out that our choice of using higher-order polynomials is not arbitrary but is guided by the principle of "truthfulness." We introduce a metric called the "truthful gap," mathematically defined as $\mathbb{T}_V(f, g)$ (as per Equation 1), to quantify how well our polynomial approximations capture the information in a specific subspace $V$ of the original function $f$. This notion of truthfulness serves as a rigorous measure to validate the efficacy of our higher-order polynomial approximations.

Our empirical results, particularly those outlined in Section 5, indicate that higher-order polynomials, specifically of orders 2 and 3, yield a favorable trade-off. They offer a substantial reduction in interpretation error and a low truthful gap compared to other techniques, substantiating the reasonableness and effectiveness of our approach.

Notice that our algorithm is a post-hoc model-agnostic interpretation algorithm, we only need to use the original neural network $f$ to infer on given input as an oracle. This means that one can easily change the network architecture without any additional changes.

# H More Experimental Results

In this section, we present ablation studies that examine the impact of various factors such as image segmentation format, neural network architecture, and choice of baselines. Our results demonstrate the robustness and efficacy of our approach in different experimental settings.

## H.1 Image Segmentation Format

We employ the SLIC algorithm to generate segmentation superpixels on the ImageNet dataset. We initially divide each $224 \times 224$ image into 16 superpixels for a balance between human readability and computational efficiency. The results, displayed in Table 6 and Figure 4, validate the effectiveness of our approach. Further experiments with 10 superpixels are reported in Table 7, confirming the robustness of our method.

## H.2 Neural Network Architectures

We also conduct experiments using diverse neural network architectures such as ResNet-18 and VGG16. Tables 8 and 9 respectively present the interpretation errors on ImageNet for these architectures.

## H.3 Choice of Baselines

The choice of baseline is critical for removal-based interpretation methods. We investigate the interpretation errors when using the average pixel value and a blurred image as baselines. The results are shown in Tables 10 and 11.

Table 6: Interpretation error on MS-COCO. Bold indicates the best performance, and underline indicates the second best.

|  | Radius | | | | |
|---|---|---|---|---|---|
|  | 1 | 2 | 4 | 8 | 16 |
| **Harmonica-2nd** | 0.0291 | 0.0332 | 0.0363 | 0.0367 | 0.0367 |
| **Harmonica-3rd** | **0.0183** | **0.0218** | **0.0241** | **0.0244** | **0.0244** |
| LIME | 0.2264 | 0.2451 | 0.2478 | 0.2479 | 0.2479 |
| SHAP | 0.1952 | 0.2036 | 0.2041 | 0.2042 | 0.2042 |
| IG | 0.3473 | 0.3515 | 0.3464 | 0.3462 | 0.3438 |
| IH | 1.3717 | 1.5047 | 1.5141 | 1.5141 | 1.5141 |
| Shapley Taylor | 0.3554 | 0.3313 | 0.3269 | 0.3267 | 0.3267 |
| Faith-Shap | 0.2428 | 0.2397 | 0.2349 | 0.2347 | 0.2347 |

Table 7: Interpretation error on ImageNet with the number of superpixels set to 10.

|  | Radius | | | | |
|---|---|---|---|---|---|
|  | 1 | 2 | 4 | 8 | 16 |
| **Harmonica-2nd** | 0.0970 | 0.1054 | 0.1221 | 0.1416 | 0.1420 |
| **Harmonica-3rd** | **0.0705** | **0.0825** | **0.0997** | **0.1179** | **0.1183** |
| LIME | 0.1995 | 0.2121 | 0.2570 | 0.3052 | 0.3051 |
| SHAP | 0.1449 | 0.2077 | 0.2743 | 0.3064 | 0.3063 |
| IG | 0.2615 | 0.3567 | 0.4423 | 0.4628 | 0.4626 |
| IH | 1.6707 | 2.2977 | 2.9443 | 3.1871 | 3.1877 |
| Shapley Taylor | 0.0982 | 0.1417 | 0.1761 | 0.1815 | 0.1814 |
| Faith-Shap | 0.0970 | 0.1250 | 0.1587 | 0.1811 | 0.1810 |

# I Detailed Results on Interpretation Error

In this section, we provide interpretation error results under other norms (Figure 6, 7, and 8) and the corresponding numerical results (Table 12, 13 and 14).

Table 8: Interpretation error on ImageNet using ResNet18 (ACC@1=69.758%).

|  | Radius | | | | |
| --- | --- | --- | --- | --- | --- |
|  | 1 | 2 | 4 | 8 | 16 |
| **Harmonica-2nd** | 0.1468 | 0.1433 | 0.1353 | 0.1357 | 0.1414 |
| **Harmonica-3rd** | 0.1247 | **0.1261** | **0.1231** | **0.1272** | **0.1319** |
| LIME | 0.2278 | 0.2209 | 0.2150 | 0.2382 | 0.2578 |
| SHAP | **0.1034** | 0.1436 | 0.1963 | 0.2323 | 0.2375 |
| IG | 0.1937 | 0.2613 | 0.3466 | 0.4028 | 0.4039 |
| IH | 0.6422 | 0.8686 | 1.1591 | 1.3939 | 1.4254 |
| Shapley Taylor | 0.1292 | 0.1530 | 0.1716 | 0.1744 | 0.1775 |
| Faith-Shap | 0.1239 | 0.1426 | 0.1532 | 0.1607 | 0.1693 |

Table 9: Interpretation error on ImageNet using VGG16 (ACC@1=71.592%).

|  | Radius | | | | |
| --- | --- | --- | --- | --- | --- |
|  | 1 | 2 | 4 | 8 | 16 |
| **Harmonica-2nd** | 0.1468 | 0.1404 | 0.1298 | 0.1303 | 0.1368 |
| **Harmonica-3rd** | 0.1290 | **0.1267** | **0.1204** | **0.1225** | **0.1289** |
| LIME | 0.2376 | 0.2280 | 0.2180 | 0.2448 | 0.2664 |
| SHAP | **0.1042** | 0.1461 | 0.2012 | 0.2407 | 0.2477 |
| IG | 0.1708 | 0.2331 | 0.3123 | 0.3659 | 0.3694 |
| IH | 0.5988 | 0.8132 | 1.1010 | 1.3663 | 1.4149 |
| Shapley Taylor | 0.1155 | 0.1435 | 0.1677 | 0.1720 | 0.1728 |
| Faith-Shap | 0.1085 | 0.1306 | 0.1456 | 0.1548 | 0.1621 |

Table 10: Interpretation error on ImageNet using the average pixel value as the baseline.

|  | Radius | | | | |
| --- | --- | --- | --- | --- | --- |
| Method | 1 | 2 | 4 | 8 | 16 |
| **Harmonica-2nd** | 0.1341 | 0.1279 | 0.1201 | 0.1243 | 0.1304 |
| **Harmonica-3rd** | **0.1012** | **0.1025** | **0.1022** | **0.1095** | **0.1156** |
| LIME | 0.2205 | 0.2142 | 0.2095 | 0.2327 | 0.2520 |
| SHAP | 0.1022 | 0.1410 | 0.1934 | 0.2307 | 0.2357 |
| IG | 0.1824 | 0.2468 | 0.3280 | 0.3822 | 0.3835 |
| IH | 0.6292 | 0.8497 | 1.1356 | 1.3752 | 1.4115 |
| Shapley Taylor | 0.1293 | 0.1665 | 0.2157 | 0.2550 | 0.2534 |
| Faith-Shap | 0.1219 | 0.1530 | 0.1918 | 0.2288 | 0.2302 |

Table 11: Interpretation error on ImageNet using the blurred image as the baseline.

|  | Radius | | | | |
| --- | --- | --- | --- | --- | --- |
| Method | 1 | 2 | 4 | 8 | 16 |
| **Harmonica-2nd** | 0.0207 | 0.0209 | 0.0210 | 0.0218 | 0.0225 |
| **Harmonica-3rd** | **0.0200** | **0.0203** | **0.0204** | **0.0213** | **0.0220** |
| LIME | 0.0260 | 0.0276 | 0.0310 | 0.0372 | 0.0394 |
| SHAP | 0.0184 | 0.0258 | 0.0360 | 0.0460 | 0.0477 |
| IG | 0.0215 | 0.0299 | 0.0415 | 0.0524 | 0.0541 |
| IH | 0.0268 | 0.0373 | 0.0525 | 0.0684 | 0.0718 |
| Shapley Taylor | 0.0350 | 0.0370 | 0.0385 | 0.0372 | 0.0360 |
| Faith-Shap | 0.0343 | 0.0358 | 0.0368 | 0.0358 | 0.0350 |

| Radius | $L^2$ norm | $L^1$ norm | $L^0$ norm | Radius | $L^2$ norm | $L^1$ norm | $L^0$ norm | Radius | $L^2$ norm | $L^1$ norm | $L^0$ norm |
|---|---|---|---|---|---|---|---|---|---|---|---|
| 1 | 0.0536 | 0.0460 | 0.1154 | 1 | **0.0434** | 0.0363 | 0.0995 | 1 | 0.0981 | 0.0870 | 0.3572 |
| 2 | 0.0576 | 0.0463 | 0.1219 | 2 | **0.0484** | 0.0376 | 0.1046 | 2 | 0.1010 | 0.0861 | 0.3458 |
| 4 | 0.0639 | 0.0483 | 0.1350 | 4 | **0.0561** | 0.0405 | 0.1157 | 4 | 0.1043 | 0.0863 | 0.3421 |
| 8 | 0.0768 | 0.0549 | 0.1696 | 8 | **0.0700** | 0.0474 | 0.1456 | 8 | 0.1135 | 0.0919 | 0.3659 |
| 16 | 0.0945 | 0.0651 | 0.2173 | 16 | **0.0876** | 0.0575 | 0.1900 | 16 | 0.1285 | 0.1015 | 0.4034 |
| 32 | 0.0990 | 0.0677 | 0.2275 | 32 | **0.0921** | 0.0600 | 0.2003 | 32 | 0.1323 | 0.1038 | 0.4111 |
| ∞ | 0.0991 | 0.0677 | 0.2279 | ∞ | **0.0922** | 0.0601 | 0.2005 | ∞ | 0.1322 | 0.1037 | 0.4112 |
| | **Harmonica**[2] | | | | **Harmonica**[3] | | | | LIME | | |

| Radius | $L^2$ norm | $L^1$ norm | $L^0$ norm | Radius | $L^2$ norm | $L^1$ norm | $L^0$ norm | Radius | $L^2$ norm | $L^1$ norm | $L^0$ norm |
|---|---|---|---|---|---|---|---|---|---|---|---|
| 1 | 0.0792 | 0.0568 | 0.1819 | 1 | 0.1133 | 0.0681 | 0.2010 | 1 | 0.0865 | 0.0477 | 0.1215 |
| 2 | 0.1089 | 0.0836 | 0.3167 | 2 | 0.1551 | 0.1055 | 0.3361 | 2 | 0.1232 | 0.0756 | 0.2118 |
| 4 | 0.1470 | 0.1186 | 0.4847 | 4 | 0.2081 | 0.1563 | 0.5066 | 4 | 0.1753 | 0.1183 | 0.3406 |
| 8 | 0.1865 | 0.1554 | 0.6252 | 8 | 0.2595 | 0.2075 | 0.6476 | 8 | 0.2343 | 0.1691 | 0.4859 |
| 16 | 0.2106 | 0.1783 | 0.6891 | 16 | 0.2852 | 0.2336 | 0.7058 | 16 | 0.2678 | 0.1994 | 0.5724 |
| 32 | 0.2137 | 0.1814 | 0.6958 | 32 | 0.2872 | 0.2357 | 0.7099 | 32 | 0.2709 | 0.2028 | 0.5823 |
| ∞ | 0.2137 | 0.1814 | 0.6961 | ∞ | 0.2874 | 0.2359 | 0.7105 | ∞ | 0.2711 | 0.2029 | 0.5823 |
| | SHAP | | | | Integrated Gradients | | | | Integrated Hessians | | |

| Radius | $L^2$ norm | $L^1$ norm | $L^0$ norm | Radius | $L^2$ norm | $L^1$ norm | $L^0$ norm |
|---|---|---|---|---|---|---|---|
| 1 | 0.0623 | 0.0472 | 0.1090 | 1 | 0.0602 | 0.0451 | 0.1034 |
| 2 | 0.0853 | 0.0649 | 0.1971 | 2 | 0.0813 | 0.0614 | 0.1795 |
| 4 | 0.1144 | 0.0874 | 0.3069 | 4 | 0.1091 | 0.0824 | 0.2794 |
| 8 | 0.1426 | 0.1081 | 0.3909 | 8 | 0.1387 | 0.1042 | 0.3707 |
| 16 | 0.1612 | 0.1217 | 0.4404 | 16 | 0.1586 | 0.1190 | 0.4271 |
| 32 | 0.1642 | 0.1241 | 0.4486 | 32 | 0.1616 | 0.1213 | 0.4356 |
| ∞ | 0.1642 | 0.1240 | 0.4489 | ∞ | 0.1616 | 0.1213 | 0.4357 |
| | Shapley Taylor Index[2] | | | | Faith-Shap[2] | | |

Table 12: The interpretation error of Harmonica and other baseline algorithms evaluated on the SST-2 dataset for different neighborhoods with a radius ranging from 1 to ∞ under $L^2$, $L^1$ and $L^0$ norms.

| Radius | $L^2$ norm | $L^1$ norm | $L^0$ norm | Radius | $L^2$ norm | $L^1$ norm | $L^0$ norm | Radius | $L^2$ norm | $L^1$ norm | $L^0$ norm |
|---|---|---|---|---|---|---|---|---|---|---|---|
| 1 | 0.0225 | 0.0193 | 0.0041 | 1 | **0.0175** | 0.0149 | 0.0027 | 1 | 0.0624 | 0.0525 | 0.1175 |
| 2 | 0.0234 | 0.0193 | 0.0036 | 2 | **0.0193** | 0.0157 | 0.0022 | 2 | 0.0671 | 0.0546 | 0.1358 |
| 4 | 0.0300 | 0.0228 | 0.0141 | 4 | **0.0264** | 0.0195 | 0.0098 | 4 | 0.0740 | 0.0583 | 0.1606 |
| 8 | 0.0442 | 0.0298 | 0.0434 | 8 | **0.0407** | 0.0265 | 0.0367 | 8 | 0.084 | 0.0640 | 0.1903 |
| 16 | 0.0506 | 0.0330 | 0.0574 | 16 | **0.0472** | 0.0296 | 0.0506 | 16 | 0.0885 | 0.0664 | 0.2017 |
| 32 | 0.0515 | 0.0334 | 0.0596 | 32 | **0.0481** | 0.0301 | 0.0528 | 32 | 0.0892 | 0.0667 | 0.2033 |
| ∞ | 0.0515 | 0.0334 | 0.0597 | ∞ | **0.0481** | 0.0301 | 0.0529 | ∞ | 0.0892 | 0.0667 | 0.2033 |
| | **Harmonica**[2] | | | | **Harmonica**[3] | | | | LIME | | |

| Radius | $L^2$ norm | $L^1$ norm | $L^0$ norm | Radius | $L^2$ norm | $L^1$ norm | $L^0$ norm | Radius | $L^2$ norm | $L^1$ norm | $L^0$ norm |
|---|---|---|---|---|---|---|---|---|---|---|---|
| 1 | 0.0849 | 0.0650 | 0.2266 | 1 | 0.1228 | 0.0810 | 0.2702 | 1 | 0.1178 | 0.0635 | 0.1929 |
| 2 | 0.1123 | 0.0903 | 0.3589 | 2 | 0.1576 | 0.1169 | 0.4102 | 2 | 0.1567 | 0.0997 | 0.3125 |
| 4 | 0.1406 | 0.1162 | 0.4891 | 4 | 0.1882 | 0.1494 | 0.5333 | 4 | 0.1969 | 0.1415 | 0.4501 |
| 8 | 0.1598 | 0.1341 | 0.5610 | 8 | 0.2031 | 0.1657 | 0.5885 | 8 | 0.2227 | 0.1702 | 0.5423 |
| 16 | 0.1662 | 0.1402 | 0.5817 | 16 | 0.2067 | 0.1697 | 0.6004 | 16 | 0.2308 | 0.1795 | 0.5702 |
| 32 | 0.1671 | 0.1411 | 0.5842 | 32 | 0.2071 | 0.1702 | 0.6014 | 32 | 0.2319 | 0.1807 | 0.5739 |
| ∞ | 0.1671 | 0.1411 | 0.5843 | ∞ | 0.2071 | 0.1702 | 0.6015 | ∞ | 0.2319 | 0.1807 | 0.5738 |
| | SHAP | | | | Integrated Gradients | | | | Integrated Hessians | | |

| Radius | $L^2$ norm | $L^1$ norm | $L^0$ norm | Radius | $L^2$ norm | $L^1$ norm | $L^0$ norm |
|---|---|---|---|---|---|---|---|
| 1 | 0.0444 | 0.0365 | 0.0554 | 1 | 0.0395 | 0.0313 | 0.0337 |
| 2 | 0.0590 | 0.0470 | 0.0999 | 2 | 0.0469 | 0.0378 | 0.0466 |
| 4 | 0.0684 | 0.0538 | 0.1339 | 4 | 0.0547 | 0.0440 | 0.0657 |
| 8 | 0.0698 | 0.0542 | 0.1300 | 8 | 0.0620 | 0.0494 | 0.0808 |
| 16 | 0.0719 | 0.0558 | 0.1287 | 16 | 0.0684 | 0.0543 | 0.0851 |
| 32 | 0.0741 | 0.0575 | 0.1294 | 32 | 0.0740 | 0.0588 | 0.0859 |
| ∞ | 0.0743 | 0.0577 | 0.1294 | ∞ | 0.0744 | 0.0590 | 0.0859 |
| | Shapley Taylor Index[2] | | | | Faith-Shap[2] | | |

Table 13: The interpretation error of Harmonica and other baseline algorithms evaluated on the IMDb dataset for different neighborhoods with a radius ranging from 1 to ∞ under $L^2$, $L^1$ and $L^0$ norms.

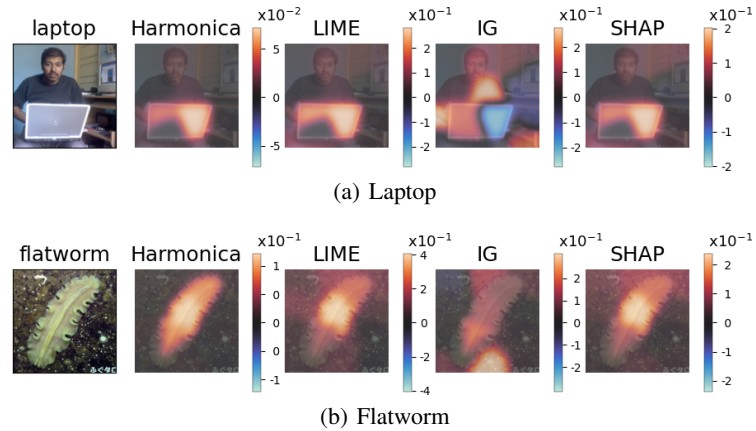

(a) Laptop

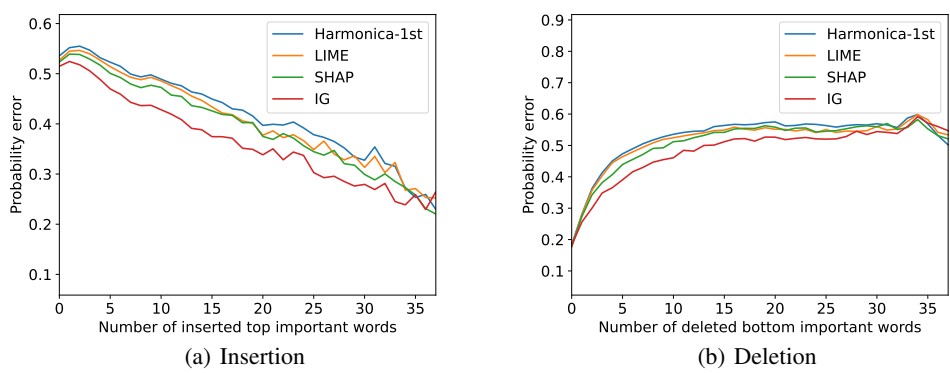

(b) Flatworm

Figure 4: Illustrative examples for applying our method to vision tasks.

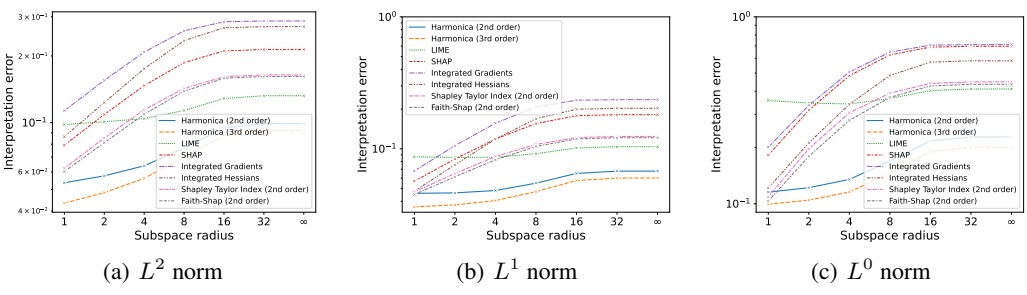

(a) Insertion

(b) Deletion

Figure 5: Insertion and deletion results on SST2.

## J  Detailed Results on Harmonica-local

### J.1  Additional Experiments

We further explore Harmonica's performance when limited to a local space instead of the whole space. It is worth noting that $L_r$ is a regularization loss for Harmonica, ensuring the sparseness of each interpretation model $g_i$. Additionally, $L_c$ is a regularization loss penalizing the difference between interpretation models. The value of balance coefficients depends on the application scenario and can be determined by the end user.

(a) $L^2$ norm

(b) $L^1$ norm

(c) $L^0$ norm

Figure 6: Visualization of interpretation error $\mathbb{I}_{p,\mathcal{N}_x}(f,g)$ evaluated on SST-2 dataset.

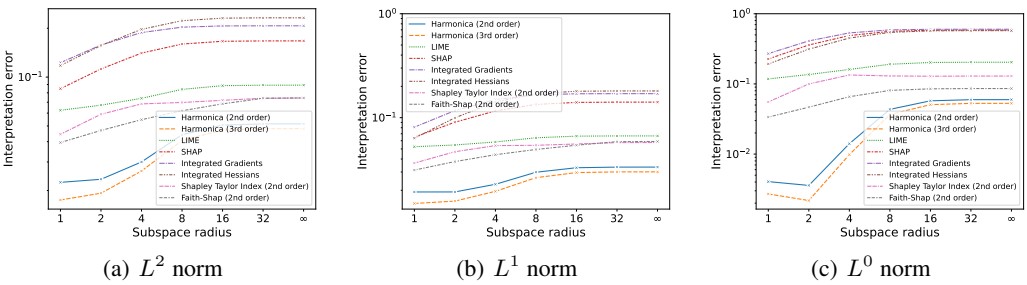

(a) $L^2$ norm      (b) $L^1$ norm      (c) $L^0$ norm

Figure 7: Visualization of interpretation error $\mathbb{I}_{p,\mathcal{N}_x}(f,g)$ evaluated on IMDb dataset.

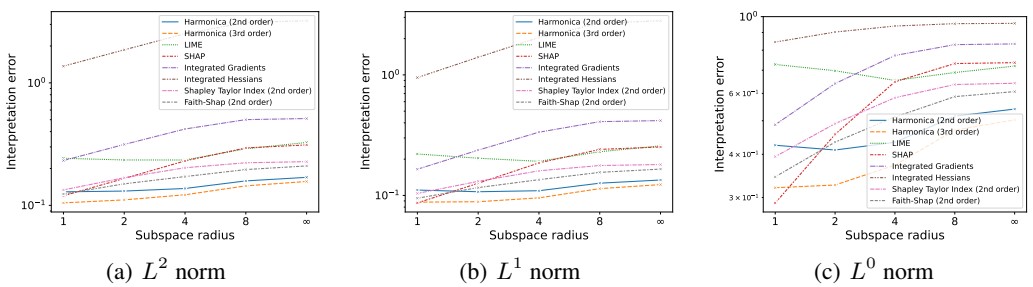

(a) $L^2$ norm      (b) $L^1$ norm      (c) $L^0$ norm

Figure 8: Visualization of interpretation error $\mathbb{I}_{p,\mathcal{N}_x}(f,g)$ evaluated on ImageNet dataset.

| Radius | $L^2$ norm | $L^1$ norm | $L^0$ norm | Radius | $L^2$ norm | $L^1$ norm | $L^0$ norm | Radius | $L^2$ norm | $L^1$ norm | $L^0$ norm |
|---|---|---|---|---|---|---|---|---|---|---|---|
| 1 | 0.1290 | 0.1108 | 0.4248 | 1 | **0.1048** | 0.0880 | 0.3202 | 1 | 0.2422 | 0.2208 | 0.7274 |
| 2 | 0.1308 | 0.1073 | 0.4116 | 2 | **0.1108** | 0.0887 | 0.3260 | 2 | 0.2347 | 0.2036 | 0.6976 |
| 4 | 0.1373 | 0.1094 | 0.4332 | 4 | **0.1220** | 0.0955 | 0.3723 | 4 | 0.2346 | 0.1918 | 0.6540 |
| 8 | 0.1584 | 0.1264 | 0.5156 | 8 | **0.1443** | 0.1139 | 0.4698 | 8 | 0.2897 | 0.2304 | 0.6893 |
| $\infty$ | 0.1693 | 0.1342 | 0.5405 | $\infty$ | **0.1566** | 0.1230 | 0.5030 | $\infty$ | 0.3261 | 0.2579 | 0.7196 |
| **Harmonica$^2$** | | | | **Harmonica$^3$** | | | | LIME | | | |

| Radius | $L^2$ norm | $L^1$ norm | $L^0$ norm | Radius | $L^2$ norm | $L^1$ norm | $L^0$ norm | Radius | $L^2$ norm | $L^1$ norm | $L^0$ norm |
|---|---|---|---|---|---|---|---|---|---|---|---|
| 1 | 0.1197 | 0.0867 | 0.2892 | 1 | 0.2322 | 0.1650 | 0.4871 | 1 | 1.3681 | 0.9504 | 0.8443 |
| 2 | 0.1658 | 0.1261 | 0.4566 | 2 | 0.3141 | 0.2375 | 0.6406 | 2 | 1.8603 | 1.4006 | 0.9025 |
| 4 | 0.2306 | 0.1862 | 0.6483 | 4 | 0.4200 | 0.3346 | 0.7722 | 4 | 2.5168 | 2.0427 | 0.9394 |
| 8 | 0.2943 | 0.2409 | 0.7318 | 8 | 0.5010 | 0.4094 | 0.8300 | 8 | 3.1095 | 2.6844 | 0.9543 |
| $\infty$ | 0.3115 | 0.2523 | 0.7362 | $\infty$ | 0.5113 | 0.4177 | 0.8340 | $\infty$ | 3.2139 | 2.8140 | 0.9562 |
| SHAP | | | | Integrated Gradients | | | | Integrated Hessians | | | |

| Radius | $L^2$ norm | $L^1$ norm | $L^0$ norm | Radius | $L^2$ norm | $L^1$ norm | $L^0$ norm |
|---|---|---|---|---|---|---|---|
| 1 | 0.1337 | 0.1039 | 0.3939 | 1 | 0.1238 | 0.0948 | 0.3443 |
| 2 | 0.1681 | 0.1309 | 0.4906 | 2 | 0.1499 | 0.1161 | 0.4338 |
| 4 | 0.2028 | 0.1600 | 0.5830 | 4 | 0.1718 | 0.1351 | 0.5126 |
| 8 | 0.2226 | 0.1771 | 0.6365 | 8 | 0.1960 | 0.1554 | 0.5872 |
| $\infty$ | 0.2274 | 0.1804 | 0.6418 | $\infty$ | 0.2099 | 0.1654 | 0.6071 |
| Shapley Taylor Index$^2$ | | | | Faith-Shap$^2$ | | | |

Table 14: The interpretation error of Harmonica and other baseline algorithms evaluated on the ImageNet dataset for different neighborhoods with a radius ranging from 1 to $\infty$ under $L^2$, $L^1$ and $L^0$ norms.

Figure 9, 10 and 11 compares Harmonica-local and Harmonica algorithms on SST-2, IMDb and ImageNet, respectively. Here we set the radius of $\mathcal{N}_x$ to be 4. The results reveal that Harmonica-local indeed performs better within the local region but fails to cover a far region.

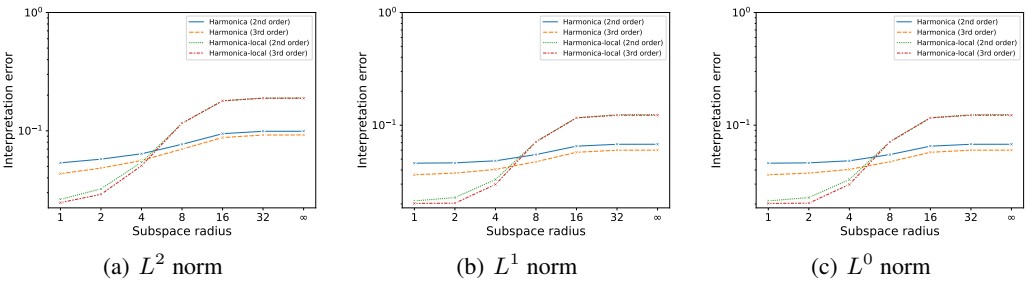

(a) $L^2$ norm  (b) $L^1$ norm  (c) $L^0$ norm

Figure 9: Visualization of interpretation error $\mathbb{I}_{p,\mathcal{N}_x}(f,g)$ evaluated on SST-2 dataset of algorithm Harmonica and Harmonica-local.

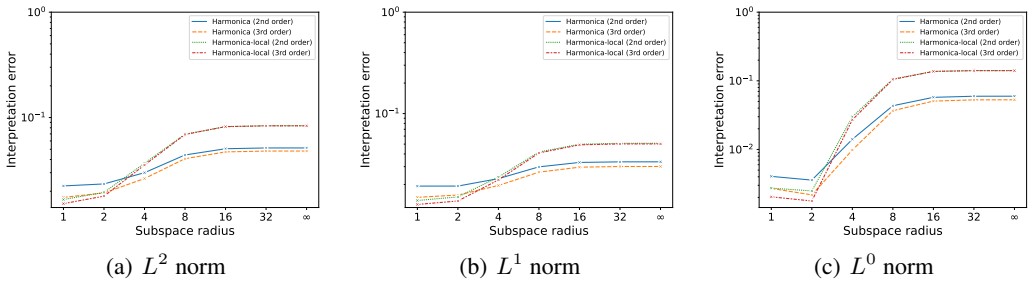

(a) $L^2$ norm  (b) $L^1$ norm  (c) $L^0$ norm

Figure 10: Visualization of interpretation error $\mathbb{I}_{p,\mathcal{N}_x}(f,g)$ evaluated on IMDb dataset of algorithm Harmonica and Harmonica-local.

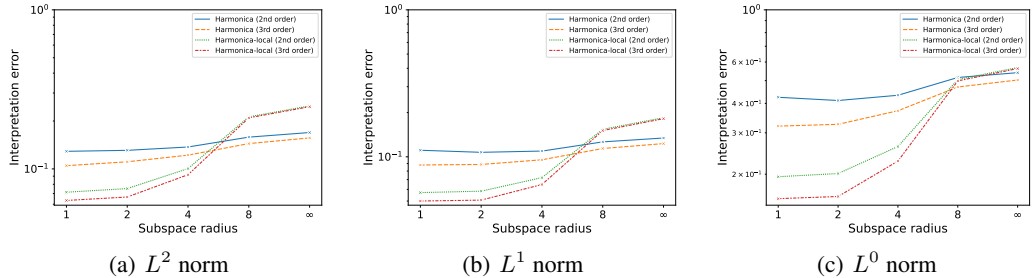

(a) $L^2$ norm  (b) $L^1$ norm  (c) $L^0$ norm

Figure 11: Visualization of interpretation error $\mathbb{I}_{p,\mathcal{N}_x}(f,g)$ evaluated on ImageNet of algorithm Harmonica and Harmonica-local.

## K  Detailed Results on Truthful Gap

**Estimating truthful gap**  For convenience, we define the set of bases $C^d$ up to degree $d$ as $C^d = {\chi_S | S \subseteq [n], |S| \leq d}$. We evaluate the truthful gap on the set of bases $C^3$, $C^2$, and $C^1$. By definition in Eqn. (3), we have $\mathbb{T}_{V_C}(f,g) = \sum_{\chi_S \in C} \langle f - g, \chi_S \rangle^2 = \left( \mathbb{E}_{x \sim \{-1,1\}^n} \left[ (f(x) - g(x)) \sum_{\chi_S \in C} \chi_S(x) \right] \right)^2$.

Then we perform a sampling-based estimation of the truthful gap. Worth mentioning that since the size of set $C^d$ satisfies $|C^d| = \sum_{i=0}^{d} \binom{n}{i}$ and the max number of words, sentences, or superpixels

$n^* \leq 50$, $\sum_{\chi_S \in C} \chi_S(x)$, as the summation function of orthonormal basis, is easy to compute on every sample $x \in \{-1, 1\}^n$ (for $n^*$ very large, we will perform another sampling step on this function).

Table 1 shows the $C^3$ truthful gap evaluated on different datasets. We can see that Harmonica outperforms all the other baseline algorithms. Figure 12 shows the truthful gap evaluated on the SST-2 dataset. We can see that Harmonica achieves the best performance for $C^2$ and $C^3$. For the simple linear case $C^1$, Harmonica is almost as good as LIME. Figure 13 shows the truthful gap evaluated on the IMDb dataset under the same settings as the SST-2 dataset. Figure 14 shows the truthful gap evaluated on the ImageNet dataset. We can see that Harmonica outperforms all the other baselines consistently.

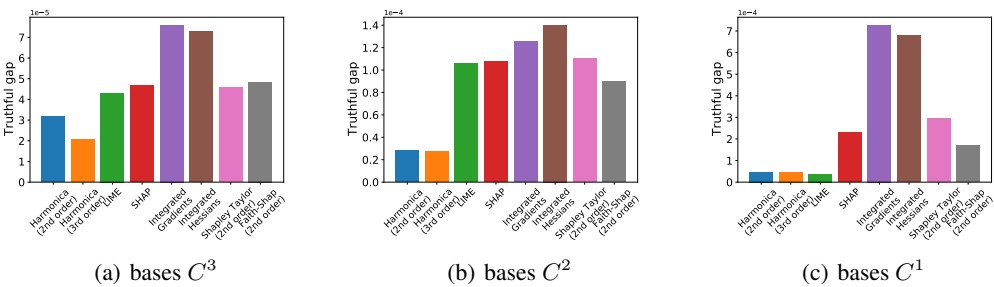

(a) bases $C^3$     (b) bases $C^2$     (c) bases $C^1$

Figure 12: Visualization of truthful gap $\mathbb{T}_C(f, g)$ evaluated on SST-2 dataset.

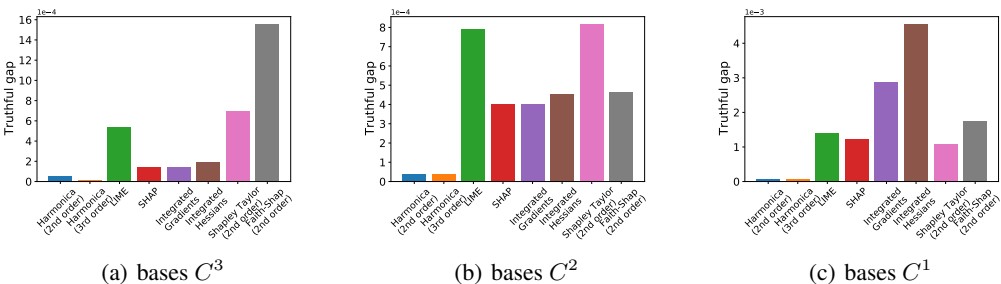

(a) bases $C^3$     (b) bases $C^2$     (c) bases $C^1$

Figure 13: Visualization of truthful gap $\mathbb{T}_C(f, g)$ evaluated on IMDb dataset.

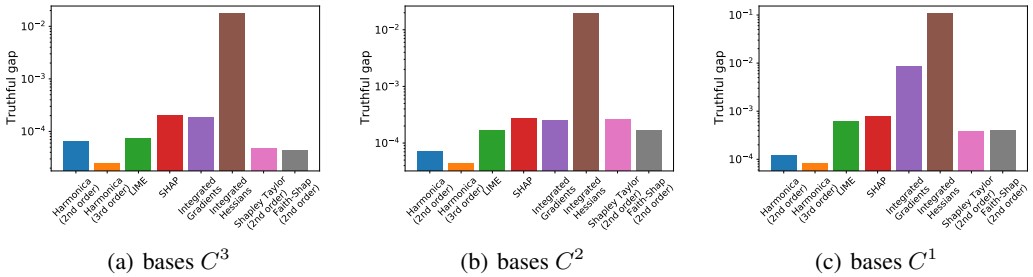

(a) bases $C^3$     (b) bases $C^2$     (c) bases $C^1$

Figure 14: Visualization of truthful gap $\mathbb{T}_C(f, g)$ evaluated on ImageNet dataset.

## L  Detailed Results on Harmonica-anchor

### L.1  Harmonica-anchor

Figure 15, 16, and 17 show the interpretation error results evaluated on three datasets. All Harmonica-anchor algorithms further reduce the interpretation error compared to Harmonica. As we increase the number of anchors, the interpretation error slightly reduces, which is consistent with our intuition. Table 15, 16, and 17 show the numerical results.

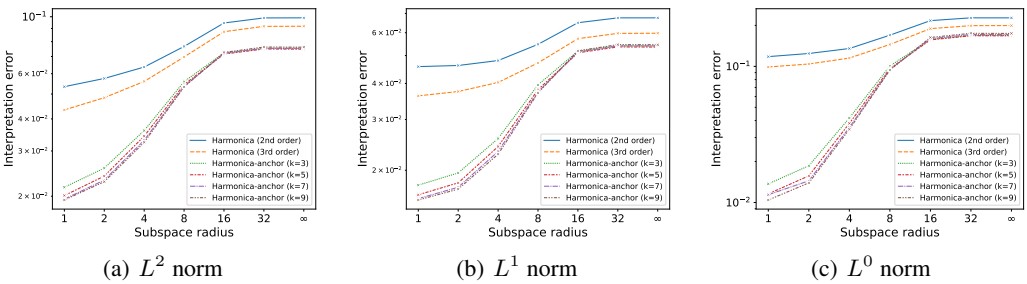

(a) $L^2$ norm  (b) $L^1$ norm  (c) $L^0$ norm

Figure 15: Visualization of interpretation error $\mathbb{I}_{p, \mathcal{N}_x}(f, g)$ evaluated on SST-2 of Harmonica-anchor (2nd order) with different anchor number $k$.

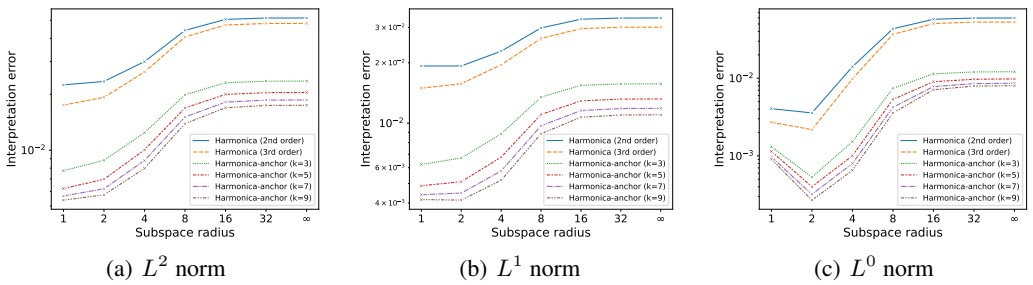

(a) $L^2$ norm  (b) $L^1$ norm  (c) $L^0$ norm

Figure 16: Visualization of interpretation error $\mathbb{I}_{p, \mathcal{N}_x}(f, g)$ evaluated on IMDb of Harmonica-anchor (2nd order) with different anchor number $k$.

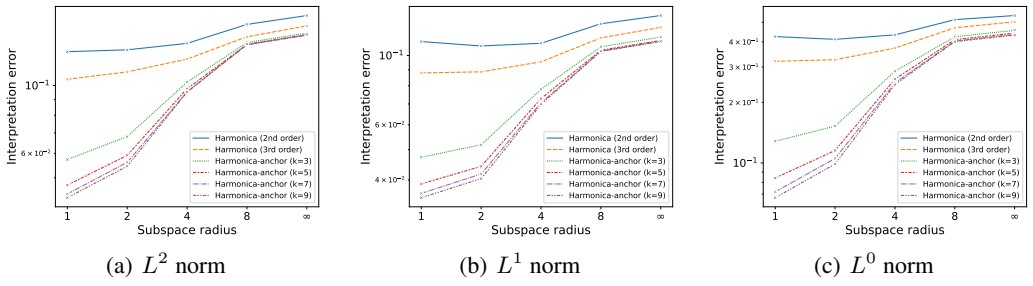

(a) $L^2$ norm  (b) $L^1$ norm  (c) $L^0$ norm

Figure 17: Visualization of interpretation error $\mathbb{I}_{p, \mathcal{N}_x}(f, g)$ evaluated on ImageNet of Harmonica-anchor (2nd order) with different anchor number $k$.

### L.2  Harmonica-anchor-constrained

Here we provide the interpretation error of Harmonica-anchor-constrained evaluated on SST-2 dataset in Figure 18. As we increase the constraint coefficient $\lambda_2$, the interpretation error increases, but still smaller than that of Harmonica.

| Radius | $L^2$ norm | $L^1$ norm | $L^0$ norm | Radius | $L^2$ norm | $L^1$ norm | $L^0$ norm |
|---|---|---|---|---|---|---|---|
| 1 | 0.0217 | 0.0178 | 0.0137 | 1 | 0.0201 | 0.0164 | 0.0116 |
| 2 | 0.0257 | 0.0196 | 0.0186 | 2 | 0.0239 | 0.0181 | 0.0157 |
| 4 | 0.0360 | 0.0258 | 0.0417 | 4 | 0.0343 | 0.0242 | 0.0381 |
| 8 | 0.0559 | 0.0396 | 0.1007 | 8 | 0.0544 | 0.0381 | 0.0963 |
| 16 | 0.0723 | 0.0518 | 0.1579 | 16 | 0.0715 | 0.0511 | 0.1565 |
| 32 | 0.0753 | 0.0541 | 0.1688 | 32 | 0.0749 | 0.0536 | 0.1678 |
| $\infty$ | 0.0753 | 0.0541 | 0.1687 | $\infty$ | 0.0748 | 0.0535 | 0.1677 |
| Harmonica-anchor ($k=3$) | | | | Harmonica-anchor ($k=5$) | | | |

| Radius | $L^2$ norm | $L^1$ norm | $L^0$ norm | Radius | $L^2$ norm | $L^1$ norm | $L^0$ norm |
|---|---|---|---|---|---|---|---|
| 1 | 0.0195 | 0.0159 | 0.0114 | 1 | 0.0193 | 0.0158 | 0.0104 |
| 2 | 0.0231 | 0.0175 | 0.0144 | 2 | 0.0228 | 0.0172 | 0.0139 |
| 4 | 0.0330 | 0.0232 | 0.0356 | 4 | 0.0324 | 0.0227 | 0.0345 |
| 8 | 0.0535 | 0.0374 | 0.0948 | 8 | 0.0533 | 0.0371 | 0.0940 |
| 16 | 0.0719 | 0.0514 | 0.1598 | 16 | 0.0726 | 0.0519 | 0.1631 |
| 32 | 0.0753 | 0.0540 | 0.1714 | 32 | 0.0762 | 0.0546 | 0.1750 |
| $\infty$ | 0.0753 | 0.0540 | 0.1717 | $\infty$ | 0.0761 | 0.0545 | 0.1746 |
| Harmonica-anchor ($k=7$) | | | | Harmonica-anchor ($k=9$) | | | |

Table 15: The interpretation error of Harmonica-anchor algorithms evaluated on the SST-2 dataset for different $k$ with a radius ranging from 1 to $\infty$ under $L^2$, $L^1$ and $L^0$ norms.

| Radius | $L^2$ norm | $L^1$ norm | $L^0$ norm | Radius | $L^2$ norm | $L^1$ norm | $L^0$ norm |
|---|---|---|---|---|---|---|---|
| 1 | 0.0078 | 0.0062 | 0.0013 | 1 | 0.0062 | 0.0049 | 0.0011 |
| 2 | 0.0088 | 0.0067 | 0.0005 | 2 | 0.0070 | 0.0051 | 0.0004 |
| 4 | 0.0124 | 0.0088 | 0.0015 | 4 | 0.0100 | 0.0068 | 0.0010 |
| 8 | 0.0199 | 0.0135 | 0.0074 | 8 | 0.0169 | 0.0110 | 0.0053 |
| 16 | 0.0231 | 0.0154 | 0.0114 | 16 | 0.0200 | 0.0129 | 0.0090 |
| 32 | 0.0235 | 0.0157 | 0.0121 | 32 | 0.0204 | 0.0132 | 0.0097 |
| $\infty$ | 0.0236 | 0.0157 | 0.0121 | $\infty$ | 0.0204 | 0.0132 | 0.0098 |
| Harmonica-anchor ($k=3$) | | | | Harmonica-anchor ($k=5$) | | | |

| Radius | $L^2$ norm | $L^1$ norm | $L^0$ norm | Radius | $L^2$ norm | $L^1$ norm | $L^0$ norm |
|---|---|---|---|---|---|---|---|
| 1 | 0.0057 | 0.0044 | 0.0010 | 1 | 0.0054 | 0.0042 | 0.0009 |
| 2 | 0.0062 | 0.0045 | 0.0003 | 2 | 0.0057 | 0.0041 | 0.0003 |
| 4 | 0.0088 | 0.0058 | 0.0008 | 4 | 0.0080 | 0.0052 | 0.0007 |
| 8 | 0.0151 | 0.0097 | 0.0042 | 8 | 0.0139 | 0.0089 | 0.0036 |
| 16 | 0.0181 | 0.0116 | 0.0078 | 16 | 0.0169 | 0.0107 | 0.0071 |
| 32 | 0.0186 | 0.0118 | 0.0085 | 32 | 0.0174 | 0.0110 | 0.0079 |
| $\infty$ | 0.0186 | 0.0119 | 0.0086 | $\infty$ | 0.0174 | 0.0110 | 0.0080 |
| Harmonica-anchor ($k=7$) | | | | Harmonica-anchor ($k=9$) | | | |

Table 16: The interpretation error of Harmonica-anchor algorithms evaluated on the IMDb dataset for different $k$ with a radius ranging from 1 to $\infty$ under $L^2$, $L^1$ and $L^0$ norms.

| Radius | $L^2$ norm | $L^1$ norm | $L^0$ norm | Radius | $L^2$ norm | $L^1$ norm | $L^0$ norm |
|---|---|---|---|---|---|---|---|
| 1 | 0.0573 | 0.0473 | 0.1281 | 1 | 0.0473 | 0.0389 | 0.0842 |
| 2 | 0.0681 | 0.0518 | 0.1524 | 2 | 0.0592 | 0.0442 | 0.1153 |
| 4 | 0.1025 | 0.0779 | 0.2855 | 4 | 0.0976 | 0.0728 | 0.2608 |
| 8 | 0.1382 | 0.1066 | 0.4240 | 8 | 0.1361 | 0.1040 | 0.4104 |
| $\infty$ | 0.1485 | 0.1145 | 0.4562 | $\infty$ | 0.1462 | 0.1118 | 0.4422 |
| Harmonica-anchor ($k=3$) | | | | Harmonica-anchor ($k=5$) | | | |

| Radius | $L^2$ norm | $L^1$ norm | $L^0$ norm | Radius | $L^2$ norm | $L^1$ norm | $L^0$ norm |
|---|---|---|---|---|---|---|---|
| 1 | 0.0442 | 0.0362 | 0.0719 | 1 | 0.0430 | 0.0351 | 0.0672 |
| 2 | 0.0561 | 0.0418 | 0.1053 | 2 | 0.0545 | 0.0404 | 0.0988 |
| 4 | 0.0954 | 0.0706 | 0.2496 | 4 | 0.0952 | 0.0698 | 0.2452 |
| 8 | 0.1358 | 0.1032 | 0.4036 | 8 | 0.1365 | 0.1032 | 0.4011 |
| $\infty$ | 0.1461 | 0.1110 | 0.4352 | $\infty$ | 0.1468 | 0.1110 | 0.4320 |
| Harmonica-anchor ($k=7$) | | | | Harmonica-anchor ($k=9$) | | | |

Table 17: The interpretation error of Harmonica-anchor algorithms evaluated on the ImageNet dataset for different $k$ with a radius ranging from 1 to $\infty$ under $L^2$, $L^1$ and $L^0$ norms.

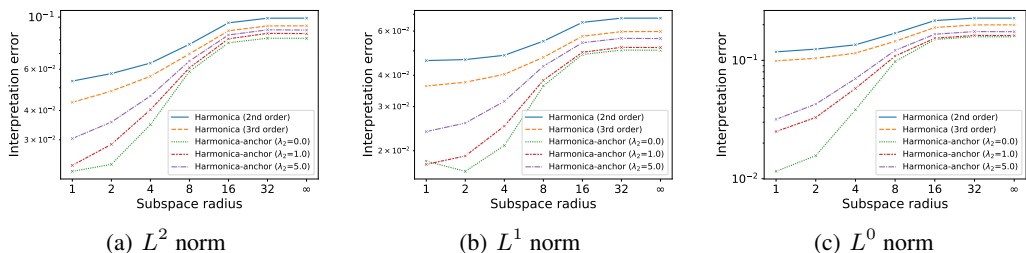

|                | (a) $L^2$ norm | (b) $L^1$ norm | (c) $L^0$ norm |

Figure 18: Visualization of interpretation error $\mathbb{I}_{p,\mathcal{N}_x}(f,g)$ evaluated on SST-2 dataset of Harmonica-anchor-constrained (2nd order) with different coefficient $\lambda_2$. Here all the Harmonica-anchor-constrained algorithms use $k = 5$.

| Radius | $L^2$ norm | $L^1$ norm | $L^0$ norm | Radius | $L^2$ norm | $L^1$ norm | $L^0$ norm |
|--------|-----------|-----------|-----------|--------|-----------|-----------|-----------|
| 1 | 0.0220 | 0.0182 | 0.0115 | 1 | 0.0234 | 0.0176 | 0.0250 |
| 2 | 0.0236 | 0.0166 | 0.0157 | 2 | 0.0287 | 0.0191 | 0.0329 |
| 4 | 0.0348 | 0.0210 | 0.0382 | 4 | 0.0403 | 0.0251 | 0.0579 |
| 8 | 0.0587 | 0.0363 | 0.0977 | 8 | 0.0608 | 0.0382 | 0.1086 |
| 16 | 0.0777 | 0.0484 | 0.1500 | 16 | 0.0808 | 0.0494 | 0.1538 |
| 32 | 0.0813 | 0.0504 | 0.1581 | 32 | 0.0853 | 0.0517 | 0.1623 |
| $\infty$ | 0.0812 | 0.0503 | 0.1579 | $\infty$ | 0.0851 | 0.0516 | 0.1620 |

Harmonica-anchor-constrained ($\lambda_2 = 0$)     Harmonica-anchor-constrained ($\lambda_2 = 1.0$)

| Radius | $L^2$ norm | $L^1$ norm | $L^0$ norm |
|--------|-----------|-----------|-----------|
| 1 | 0.0304 | 0.0238 | 0.0317 |
| 2 | 0.0357 | 0.0258 | 0.0427 |
| 4 | 0.0461 | 0.0315 | 0.0701 |
| 8 | 0.0650 | 0.0435 | 0.1214 |
| 16 | 0.0839 | 0.0539 | 0.1663 |
| 32 | 0.0883 | 0.0561 | 0.1750 |
| $\infty$ | 0.0881 | 0.0560 | 0.1745 |

Harmonica-anchor-constrained ($\lambda_2 = 5.0$)

Table 18: The interpretation error of Harmonica-anchor-constrained algorithms evaluated on the SST-2 dataset for different $\lambda_2$ with a radius ranging from 1 to $\infty$ under $L^2$, $L^1$ and $L^0$ norms.

# M  Discussion on the Low-degree algorithm

We also investigate the sample complexity of Harmonica and Low-degree algorithms, which demonstrates that Harmonica achieves better performance with the same sample size.

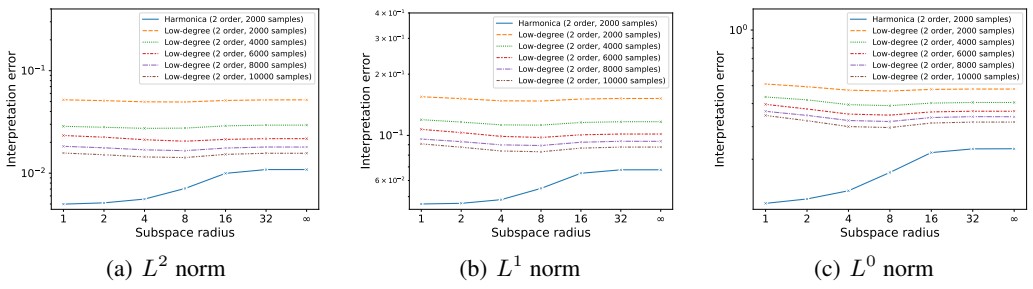

|                | (a) $L^2$ norm | (b) $L^1$ norm | (c) $L^0$ norm |

Figure 19: Visualization of interpretation error $\mathbb{I}_{p,\mathcal{N}_x}(f,g)$ evaluated on SST-2 dataset, while Harmonica and Low-degree algorithms using different sample size varying from 2000 to 10000.

From Theorem 2 and Theorem D.1, we know that the sample complexity of the Harmonica algorithm ($\tilde{O}(\frac{1}{\epsilon})$) is much more efficient than the Low-degree algorithm ($\tilde{O}(\frac{1}{\epsilon^2})$). Figure 19 shows that when evaluating the interpretation error on SST-2 dataset, with the same sample size, the Harmonica

algorithm outperforms the Low-degree algorithm by a large margin. We further increase the sample size for the Low-degree algorithm and see that its interpretation error gradually approaches that of Harmonica. However, even with 5x sample size, the Low-degree algorithm still gives a larger interpretation error compared with Harmonica. However, since exactly computing the Low-degree algorithm is extremely time-consuming, here we present the results using five times the sample size to show the calculation difficulty of the Low-degree algorithm.

