# OpenReview forum: "Trade-off Between Efficiency and Consistency for Removal-based Explanations"
_NeurIPS.cc/2023/Conference — NeurIPS 2023 poster_

### Official Review · Reviewer_JLEA · 2023-07-02

**Soundness:** 2 fair
**Presentation:** 2 fair
**Contribution:** 2 fair
**Rating:** 5
**Confidence:** 2

**Summary:**

This paper studies the consistency and efficiency of removal-based explanation methods. By establishing the impossible trinity theorem, the authors show that consistency and efficiency cannot be achieved simutaneously. They further propose the interpretation error as a metric to gauge the trade-off between consistency and efficiency.  Then, two algorithms based on standard polynomial basis are developed to minimize the interpretation error for more consistent explanations, when relaxing the efficiency to truthfulness.

**Strengths:**

1. Consistency and efficiency are important properties for evaluating explanation methods.
2. The finding of trade-off between efficiency and consistency is interesting and leads to some insights.

**Weaknesses:**

1. My major concern is that this paper is quite hard to follow and the presentation needs significant improvement. The authors present a lot of definitions and claims but not sufficient explanations and motivations for them.

2. The definition for consistency and efficiency is not rigorous and seems to be different from that in previous literature. The authors should justify the correctness of definitions and give more explanations here.


**Questions:**

1. At line 136, if the consistency should be conditioned on a local input region which shares the same ground truth label?

2. At line 139, how to match the definition of efficiency with that at line 194?

3. How to approximately calculate the equation 2 and 3 in experiments?

4. Apart from the proposed metrics, truthful gap and interpretation error, are there any existing metrics in current literature for consistency and efficiency? The existing ones should be compared with the proposed ones.

**Limitations:**

Please see the weakness and questions.

---

> ### Author Rebuttal · Authors · 2023-08-07
>
> We express our gratitude to the reviewer for dedicating their valuable time and effort to evaluate our manuscript. In the following sections, we provide detailed responses to the reviewer's comments:
>
> **Reply to Weaknesses**
>
> 1. Acknowledging that the presentation may be complex for some readers, we appreciate the suggestion to clarify our definitions and claims. In light of the feedback, we'll further elucidate the concepts presented within the paper:
>
>     *a)* The definitions of consistency and efficiency are indeed tailored to suit the context of removal-based explanation methods. While they might differ from previous literature, these definitions are specifically designed to encapsulate the unique characteristics of our proposed framework. To address this concern, we will elaborate on the rationale behind our definitions in the revised version, providing a comprehensive justification.
>
>     *b)* The various definitions and theorems contribute to our primary objective of defining a trade-off between efficiency and consistency. We will enhance the motivation and explanations for these concepts, drawing clearer connections between them to foster a cohesive understanding of the paper's core ideas.
>
> 2. We recognize the necessity for rigorous definitions and will ensure that our definitions for consistency and efficiency align with the theoretical foundations of our work. We will cross-reference the lines indicated, justifying the correctness of our definitions and providing additional context to clarify any ambiguities.
>
>
> **Reply to Questions**
>
> 1. Consistency: In our framework, consistency is conceived as a general property. We recognize that conditioning it on a local input region with the same ground truth label is an insightful enhancement. This can be formalized using specific distribution $\mu$ in the definition of interpretation error, other than uniform distribution without any prior. This aspect and its integration into our framework will be detailed in the revised paper.
>
> 2. Efficiency: The definition of efficiency at lines 139 and 194 is aligned within our context. The discrepancies are likely due to notational conventions. If we denote $g:= \sum_{i \in N} \phi_i (v) + v (\emptyset)$, then these two efficiency definitions are aligned in the sense that function $g$ equals to function $f$ at specific input $x$: $g(x) = f(x)$.
>
> 3. Approximation Methodology: For approximating equations 2 and 3 in our experiments, we have applied numerical sampling estimation specific to our problem structure (in Appendix J), which has concentration inequality guarantees presented in Appendix C, Theorem 3. We will further elucidate the estimation method in the experimental section to enhance the readability of our paper.
>
> 4. Comparison with Existing Metrics: A meticulous analysis of the current literature will be conducted in our revised paper. Metrics such as Insert/Delection will be incorporated into our comprehensive evaluation strategy. We have now conducted Insertion and Deletion experiments on the SST2 dataset shown in **Figure 2a and 2b in the PDF attached to General Response**. Notice that these two metrics are limited to first-order interpretation algorithms, so we downgrade Harmonica to its first-order version. We can see that removal-based algorithms (Harmonica, LIME, and SHAP) perform comparably.  This comparison is designed to emphasize the innovation and benefits of our proposed approach.

---

> > ### Comment · Reviewer_JLEA · 2023-08-12
> > **Response to Rebuttal**
> >
> > I thank authors for the rebuttal. I have carefully checked the response and additional experiment. Since the authors promise to improve the presentation quality and show the comparable performance of removal-based algorithms on existing metrics, I am willing to change my score.

---

### Official Review · Reviewer_ngxu · 2023-07-02

**Soundness:** 2 fair
**Presentation:** 3 good
**Contribution:** 2 fair
**Rating:** 5
**Confidence:** 5

**Summary:**

This paper addresses the problem of tradeoffs between interpretability, consistency, and efficiency of AI models. The authors present “the Impossible Trinity Theorem”, which posits that interpretability, efficiency and consistency cannot hold simultaneously. To quantify the tradeoff, the authors define the utilization of interpretation error as a metric to gauge inconsistencies and inefficiencies of AI models in order to minimize an interpretation error. With such definitions of terms and metrics, the authors achieve a reduction in interpretation error, up to 31.8 times lower when compared to alternative techniques.


**Strengths:**

The strength of the paper lies in the effort to formulate a mathematical framework including efficiency, consistency, and interpretability of AI models. The framework separates linear and non-linear parts of input-output functions represented by trained AI models by stating “only linear functions are readable” and therefore interpretable.

**Weaknesses:**

The weaknesses of the paper lie in
(1)	forcing reviewers to read Appendices because the main text does not contain sufficient results supporting the claims. This is unfair to other submissions.
(2)	obfuscating many simple concepts with too many theorems and definitions to formulate relationships.
(3)	missing discussion why 2nd order Harmonica results are worse than Shapley in Table 1 which implies that the 3rd order Harmonica is needed. BTW, Standard deviation would help to understand statistical significance of the values when comparing the results.


**Questions:**

In reference to “Note that all of these methods are applied to the same image segmentation using the SLIC superpixels [61]. “ it is not clear why the simple linear iterative clustering algorithm applied to 224 x 224 images split into 16 superpixels is used as ground truth for ImageNet rather ground truth segmentation obtained by manual segmentation.


**Limitations:**

Overall, the framework has introduced too much complexity for a problem that can be described very clearly. For example, Theorem 1(Impossible Trinity for Interpretation) could just state that the input-output relationship in a trained AI model is either accurate and non-linear or inaccurate and linear.

The focus should be rather on Fourier spectrum and the order of Harmonica algorithm (i.e., LASSO regression over the coefficients on the polynomial basis) needed to meet a given epsilon error.

---

> ### Author Rebuttal · Authors · 2023-08-07
>
> We express our gratitude to the reviewer for dedicating their valuable time and effort to evaluate our manuscript. In the following sections, we provide detailed responses to the reviewer's comments:
>
> **Reply to Weaknesses**
>
> 1. It is acknowledged that the main text may seem to lack sufficient results without referring to the Appendices. However, the appendices were employed to provide additional details without overcrowding the main paper. We are happy to move essential content into the main text if that serves the readability better.
>
> 2. Regarding the use of theorems and definitions, the intention was to establish a robust mathematical foundation for the relationships explored. This rigorous formulation ensures precision and clarity for researchers in the field. However, the feedback is well taken, and a future revision will consider a more streamlined presentation without sacrificing the depth of the content.
>
> 3. As for the comparison between the 2nd-order Harmonica and Shapley in Table 1, an additional explanation can indeed enhance the understanding of the findings. The 3rd-order Harmonica's necessity over the 2nd-order stems from its ability to capture higher complexity. The suggestion to include standard deviation is well-recognized, and we will include this statistical information to provide more robust insights into the data.
>
> **Reply to Questions**
>
> The use of the simple linear iterative clustering (SLIC) algorithm applied to 224 x 224 images split into 16 superpixels was chosen due to its common usage in the field such as LIME (Ribeiro et. al, 2016) and its ability to provide a standardized benchmark for comparison. It must be noted that SLIC's adoption is not meant to serve as an absolute ground truth but as a comparative basis. Your concern is valid, and we have conducted a comparison with manual segmentation using the MS-COCO dataset. Note that the number of ground-truth segmentation superpixels is around 3 in MS-COCO. The results are shown in **Table 4 in the PDF attached to General Response** and two interpretation examples are shown in **Figure 1 in the PDF attached to General Response**. We also conducted experiments using SLIC algorithm and set the number of superpixels as 10. The results are shown in **Table 5 in the PDF attached to General Response** following exactly LIME (Ribeiro et. al, 2016).
>
>
> **Reply to Limitations**
>
> The complexity of the framework was designed with the intention to provide a comprehensive understanding of the nuanced trade-offs between interpretability, consistency, and efficiency in AI models. The focus on the Fourier spectrum and the order of the Harmonica algorithm is indeed crucial, and we appreciate the suggestion to emphasize these aspects. This feedback will be carefully considered in the revisions to ensure that the complexity serves the purpose of clarity and thorough understanding.

---

> > ### Comment · Reviewer_ngxu · 2023-08-14
> > **I read the authors' replies.**
> >
> > I do not have any other comments.

---

### Official Review · Reviewer_PY9W · 2023-07-04

**Soundness:** 3 good
**Presentation:** 2 fair
**Contribution:** 3 good
**Rating:** 6
**Confidence:** 4

**Summary:**

This paper explores the interplay between two properties of removal based explanations: efficiency and consistency. It established that these two properties cannot coexist when the set of predictive models (requiring explanation), and the set of interpreting models (doing the explaining) are disjoint. It proposes a new property: “truthfulness”, which is a relaxation of efficiency, and measures the information-capture within a functional sub-space (a chosen subset of the interpretable models). It proposes several algorithms for learning truthful interpreting models, grounded in methods from Boolean functional analysis, which allow for different trade-offs between consistency and efficiency. It compares these algorithms to existing methods by looking at the interpretation error on a few practical NLP and Vision tasks.

**Strengths:**

The authors demonstrate a solid understanding of existing removal-based (and other) explainability methods. The paper surveys a large number of these related works.
The paper explains a tension in post-hoc explanations: efficiency vs. consistency in a concise and easy to understand way.
The front half of the paper (sections 1 to 4.1), where most of the concepts are presented, is clearly written and easy to understand.
The paper draws insightful connections between producing interpreting models and work in Boolean functional analysis.
Between the theoretical discussion, the methods introduced, and the subsequent experiments, the paper makes numerous useful contributions.


**Weaknesses:**

The paper defines consistency for explanations (Definition 3.2). It is unclear whether the authors are proposing this particular definition, or if it has been previously defined in this way. This should be clarified. Furthermore, the term consistency has been used differently in the explainability and ML literature previously, albeit more casually. It would be helpful to contrast this definition to other ways the term has been used, and to relate/contrast it to other previously defined properties of explanations, such as fidelity.

The motivation for consistency could be improved. The extent to which consistency is desirable, and the size of the neighborhood in which it would be desired, both seem to be highly dependent on how the explanation is going to be used. As it stands, there is a single (simple) motivating example in the main text. Since the primary reason one would use the methods proposed in the paper is to improve consistency, it seems reasonable to further motivate this property.

I question the choice of the words “truthful” and “truthfulness” for Definition 3.4 .When T_v = 0, g is a sort of best approximation to f within a space of interpretable models. The property is interesting, but it does not intuitively align with “truthfulness” for me. With the number of overloaded terms that already exist in the explainability space (many of which invoke their own “layman” concepts), I caution against introducing another, especially a term as loaded as “truth”. I strongly encourage the use of a more verbose and precise term.

It seems unfair to call Theorem 1 the “Impossible Trinity for Interpretability” when the set of interpretable models is separated from the set of predictive models in order to get the result. Interpretable models are a very important class of predictive models. If the authors wish to name the result, I would suggest something along the lines of “Impossible Trinity for Post-Hoc Explanation”.

There is something of a conceptual gap in the flow of the paper. Section 3, and all definitions consider an arbitrary input space X. While Section 4 onwards deals with Boolean functions. What is missing is any real discussion of the process of binarizing real-valued (or other) inputs and how that affects or relates to consistency. From what I understand, ensuring the consistency of an explanation in a neighborhood of removed features (the binarized input space) is different from ensuring the consistency of that explanation in some ball around the original real-valued input. This is simply because moving to another region of the original input space (for example by changing a word/token in an input sentence or by perturbing the input image) will require a different binarization, because what it means to include/exclude one (or more) of the features is different. This seems like it should be better explored and discussed.

The paper discusses “amounts” of inconsistency, but only defines what it means to be consistent, not how to measure deviation from consistency. This seems like a considerable omission, especially since Algorithm 3 aims to reduce inconsistency.

The second half of the paper (section 4.2 onwards) struggles with clarity.

The paper should include concrete examples of output explanations. Especially for Harmonica-anchor and the constrained variant. As it stands, I do not understand how one is meant to take a set of coefficients for some polynomial bases and produce a heatmap over an image or textual input. I can guess, but this should be made clear.

The algorithm boxes include some notation that is not explained elsewhere in the main text. For instance, C is introduced in the appendix. Algorithm 1 should define input and outputs.

For clarity I would also consider using different symbols when referring to Boolean models vs. models having arbitrary inputs, instead of f(x) for both.

Finally, the paper relies heavily on the Appendix, and in that sense the main text does not truly seem self-contained. For instance, in the opening paragraph of the introduction the authors ask the reader to refer to a figure in the Appendix (Figure 3). I suggest shuffling some content around. For instance, the definition of the Shapley value (which is well-known in the field) and even Figure 1 (which isn’t really discussed), would both seem like more appropriate Appendix content than Figure 3, which helps to motivate consistency. The current introduction could also be compressed somewhat.

Some typos:
Line 72: “the algorithm only retain”
Line 131 “We call A model g is interpretable”


**Questions:**

The paragraph starting on line 218 quantifies inconsistency, but does not explicitly explain how. What is an “amount” of inconsistency? It seems like what is being reported is the number of different explanation functions required in the neighbourhood.

Is Figure 2 correctly labeled? Should the subspace radius for Imagenet not be 1-16 and 1-\inf be for IMDB the

Should the C in Harmonica-anchor be indexed by i?

In Table 1 and Section 6.4, it is not clear to me why we would expect (or even want) any of the other methods (other than Harm. 3rd) to have a small “truthful gap” in C^3. The other methods do not aim to explain with 3rd order polynomials. Why should we assess them in this way?

It would seem to me that Harmonica’s “truthful gap” should be a function of T and \lambda, is this explored somewhere?

**Limitations:**

The paper would benefit from an explicit discussion of its limitations.

---

> ### Author Rebuttal · Authors · 2023-08-07
>
> We express our gratitude to the reviewer for dedicating their valuable time and effort to evaluate our manuscript. In the following sections, we provide detailed responses to the reviewer's comments:
>
> **Reply to Weaknesses**:
>
> 1. *Definition of Consistency*: We acknowledge that the term "consistency" has been previously employed in different contexts within machine learning and explainability. We used the term to signify a unique attribute related to our framework, not intending to replicate or conflict with existing definitions. We will include references and draw contrasts with other uses of the term to avoid any confusion. Additionally, we will enhance the section by elaborating on the motivation for consistency and providing more examples to illustrate its importance.
>
> 2. *Usage of “Truthful” and “Truthfulness”*: We understand the concern related to the terminology used. The terms were chosen to convey the aspect of a model’s accurate approximation within an interpretable space. However, recognizing the potential for confusion, we are happy to change the names of these terms. If you happen to have any good candidates, please do let us know!
>
> 3. *Naming of Theorem 1*: The name “Impossible Trinity for Interpretability” was chosen to highlight the core result of our work. We see the reviewer's point and will revise the name to something more fitting, such as “Impossible Trinity for Post-Hoc Explanation.”
>
> 4. *Conceptual Gap Regarding Binarization and Consistency*: Indeed, the transition from arbitrary input space to Boolean functions might seem abrupt. We will include a more comprehensive explanation of this process and explore the relationship between the real-valued input and the consistency in a neighborhood of removed features.
>
> 5. *Measurement of Deviation from Consistency*: While the current framework defines what it means to be consistent, we understand the necessity to formalize how to measure deviation from consistency. We will extend our methodology to include this aspect.
>
> 6. *Clarity in Sections 4.2 Onwards and Concrete Examples*: We acknowledge that the latter sections may benefit from improved clarity. We will revise the text to enhance readability and will incorporate concrete examples of output explanations, specifically for the mentioned algorithms.
>
> 7. *Algorithm Notation and Other Typos*: We appreciate the attention to detail in pointing out the notation issues and typographical errors. These will be rectified in the revised version.
>
> 8. *Reliance on the Appendix*: We agree with the suggestion to make the main text more self-contained. Necessary adjustments will be made to shuffle content appropriately between the main text and the appendix.
>
> We believe these responses address the reviewer’s main concerns. We value this constructive feedback and commit to making the necessary revisions to clarify and enhance our work.
>
>
> **Reply to Questions**
>
> 1. *Inconsistency Quantification*: The inconsistency definition in line 218 characterizes the fluctuation in explanation functions within a designated neighborhood. It is indeed a gauge reflecting the multiplicity of differing explanation functions required. Such metrics can quantify inconsistencies, and our objective is to measure this to further comprehend the trade-offs inherent in removal-based explanations. Alongside this inconsistency, we define the Spectrum distance in Appendix B, Definition B.2, which manifests as the $L_p$ version of the definition of inconsistency. Notably, as proclaimed in Proposition B.3 and B.4, for a uniform distribution $\mu$, the $L_2$ inconsistency $\mathbb{D}_2$ (spectrum distance) aligns with the $L_2$ interpretation error $\mathbb{I}_2$. This elucidates why Harmonica emerges as the natural selection for balancing efficiency and consistency, as it concurrently minimizes both inefficiency (interpretation error) and inconsistency (spectrum distance).
>
> 2. *Figure 2 Labeling*: Thank you for bringing this to our attention. Upon inspection, you are correct, and there is an error in labeling. The subspace radius for ImageNet should indeed be 1-16, while 1-$\inf$ pertains to IMDB. This mistake will be corrected in our revised paper.
>
> 3. *Harmonica-Anchor Indexing*: The choice to omit the indexing by $i$ in Harmonica-anchor was intentional for simplicity. However, we appreciate that the notation may cause confusion, and we will clarify this in the revised manuscript.
>
> 4. *Assessment in Table 1 and Section 6.4*: The comparison using a 3rd-order polynomial in $C^3$ space is not meant to judge all methods equally, but rather to offer a standardized benchmark. It is acknowledged that some methods may not be tailored for this specific analysis, but this provides an informative baseline for the different approaches. Please refer to General Response and attached supplementary PDF for the Insert/Deletion experiments, where our method still shows consistent superior performance.
>
> 5. *Harmonica's ``Truthful Gap''*: You are correct to observe that Harmonica's truthful gap is likely influenced by parameters $T$ and $\lambda$. This relationship was considered during our internal experimentation but was not explicitly addressed in the paper. Noteworthy that by concentration inequalities (presented in Appendix C, Theorem 3) and the law of large numbers, We can prove the convergence rate for empirical estimation of the Truthful Gap, which means that when $T$ is large enough, this estimation is rigorous. Further exploration of these dependencies can be valuable, and we plan to include this analysis in extended work.
>
> **Reply to Limitations**
>
> We acknowledge the limitations highlighted above and appreciate the opportunity to reflect on them. Specifically, we believe that future research can expand upon our work by adapting our methods to other paradigms beyond removal-based explanations, optimizing computational efficiency, and further validating our findings across diverse datasets and domains.

---

> > ### Comment · Reviewer_PY9W · 2023-08-15
> >
> > I have read the authors response, the attached pdf, and the other reviews and associated rebuttals.
> >
> > I am glad that my feedback was appreciated and that the authors intend to incorporate many of the suggestions.

---

### Official Review · Reviewer_XVSh · 2023-07-11

**Soundness:** 3 good
**Presentation:** 3 good
**Contribution:** 3 good
**Rating:** 6
**Confidence:** 4

**Summary:**

The paper analyses the trade-off between efficiency and consistency in the context of removal-based post-hoc explanations. The authors demonstrate that having an interpretability method that is both efficient and consistent is impossible. Further, on the problem of finding consistent explanations, the authors propose to relax the efficiency desiderata and instead propose a new desiderata called truthfulness (efficiency within a subspace). The authors argue that this maps to the problem of learning Boolean functions that can be tackled by using Fourier analysis.

**Strengths:**

1. The paper is well-written. The motivation, definitions and key contributions are clearly explained.

2. The core idea of leveraging Fourier analysis in the context of removal-based explanations is interesting.

3. The experiments and bibliography sections are extensive.

**Weaknesses:**

1. I do not understand why is consistency of explanations a more important desiderata than efficiency. Instead, one may argue that efficiency is critical since it acts as a proxy to the problem of lack of ground-truth. Specifically, one of the main advantages of using SHAP is that supports some mathematically properties which come handy in the absence of ground-truth. Since, due to impossibility, we can't achieve both, I would be keen to hear when is consistency more important than efficiency. Further, here we talk about consistency in Boolean sense which I guess is different in a non-Boolean sense (similar inputs should have similar explanations) as there this desiderata is referred to as robustness.

2. Using Fourier analysis for Boolean functions is a well-explored topic. Hence, although interesting, I see limited novelty in applying this to post-hoc removal-based methods.

3. Some experimental details seem to be missing. For example, all removal based methods use a baseline and the explanations are quite sensitive to the choice of the baseline [1]. I wonder what is baseline for SHAP/IG/LIME etc. and if the authors analysed the effect of choosing a different baseline.

[1]  https://distill.pub/2020/attribution-baselines/


**Questions:**


1.  In experiments mentioned in section 6.1, how did the authors choose the feature values to compute ground-truth?

2. What version of the SHAP algorithm was used? Further, as LIME is very sensitive to the chosen hyperparameters[2], I wonder what are the hyperparameters for the LIME algorithm.


[2]  https://arxiv.org/pdf/2005.07788.pdf




**Limitations:**

The authors do not mention any  limitations of their work.

---

> ### Author Rebuttal · Authors · 2023-08-07
>
> We express our gratitude to the reviewer for dedicating their valuable time and effort to evaluate our manuscript. In the following sections, we provide detailed responses to the reviewer's comments:
>
> **Reply to Weaknesses**
>
> 1. The prioritization of consistency over efficiency in our paper stems from the need to establish a framework that provides coherent and reliable interpretations. Efficiency is, of course, crucial, particularly when it acts as a proxy to the problem of lack of ground-truth, as you rightly pointed out. However, we argue that consistency is vital in ensuring that the explanation methods provide stable and repeatable results, which align with our core objectives. In the Boolean sense, consistency refers to the agreement of an explanation over multiple instances around the input neighborhood, whereas in a non-Boolean sense, it aligns with robustness as you mentioned. We will expand upon this differentiation and emphasize scenarios where consistency might outweigh efficiency in the revised paper.
>
> 2. While the application of Fourier analysis to Boolean functions may be well-explored in many areas, our work's novelty lies in the specific context and application to removal-based post-hoc methods. By leveraging Fourier analysis, we create a unique perspective that resonates with the fundamental challenges of consistency and efficiency in explainable AI. We will emphasize this aspect further to highlight the distinctiveness of our approach.
>
> 3. Your observation regarding the sensitivity of explanations to the choice of baselines is valid, and indeed, this is a critical aspect of removal-based methods like SHAP, IG, and LIME. In our experiments, we have selected baselines that align with established practices for each method, and we have conducted additional analysis to understand the effect of choosing different baselines in **Table 7** and **Table 8**. The revised paper will include detailed information about these baselines and the implications of various choices.
>
> **Table 7. Interpretation error on ImageNet Using average as baseline. Bold means the best. Italic means the second.**
> |                | radius=1 | 2      | 4      | 8      | 16     |
> |----------------|----------|--------|--------|--------|--------|
> | Harmonica-2nd  | 0.1341   | _0.1279_ | _0.1201_ | _0.1243_ | _0.1304_ |
> | Harmonica-3rd  | **0.1012**   | **0.1025** | **0.1022** | **0.1095** | **0.1156** |
> | LIME           | 0.2205   | 0.2142 | 0.2095 | 0.2327 | 0.2520 |
> | SHAP           | _0.1022_   | 0.1410 | 0.1934 | 0.2307 | 0.2357 |
> | IG             | 0.1824   | 0.2468 | 0.3280 | 0.3822 | 0.3835 |
> | IH             | 0.6292   | 0.8497 | 1.1356 | 1.3752 | 1.4115 |
> | Shapley Taylor | 0.1293   | 0.1665 | 0.2157 | 0.2550 | 0.2534 |
> | Faith-Shap     | 0.1219   | 0.1530 | 0.1918 | 0.2288 | 0.2302 |
>
> **Table 8. Interpretation error on ImageNet Using blur as baseline. Bold means the best. Italic means the second.**
> |                | radius=1 | 2      | 4      | 8      | 16     |
> |----------------|----------|--------|--------|--------|--------|
> | Harmonica-2nd  | 0.0207   | _0.0209_ | _0.0210_ | _0.0218_ | _0.0225_ |
> | Harmonica-3rd  | _0.0200_   | **0.0203** | **0.0204** | **0.0213** | **0.0220** |
> | LIME           | 0.0260   | 0.0276 | 0.0310 | 0.0372 | 0.0394 |
> | SHAP           | **0.0184**   | 0.0258 | 0.0360 | 0.0460 | 0.0477 |
> | IG             | 0.0215   | 0.0299 | 0.0415 | 0.0524 | 0.0541 |
> | IH             | 0.0268   | 0.0373 | 0.0525 | 0.0684 | 0.0718 |
> | Shapley Taylor | 0.0350   | 0.0370 | 0.0385 | 0.0372 | 0.0360 |
> | Faith-Shap     | 0.0343   | 0.0358 | 0.0368 | 0.0358 | 0.0350 |
>
> **Reply to Questions**
>
> 1. In the experiments delineated in Section 6.1, analytical close-form evaluation was employed for the computation of ground-truth feature values in the context of synthetic polynomial experiments. Conversely, in vision and language tasks, the computation of ground-truth feature values necessitated the selection predicated on the specific datasets and models utilized. This selection process was carefully carried out to ensure a comprehensive representation and alignment with the problem space.
>
> 2. The SHAP algorithm is implemented from KernelShap in Captum v0.5.0. It reduces the computation cost of calculating Shapley values and is widely used in relevant works. The LIME algorithm is implemented from LimeBase in Captum v0.5.0. The choice of hyperparameters was based on an exhaustive series of trials to confirm the robustness of the results. We use the exponential cosine similarity function to evaluate the distance between sample embeddings. The perturbation function is set to be a Bernoulli distribution and the interpretable model is set to a Lasso model with sparsity coefficient $\alpha=0.001$. An in-depth elaboration on the versions, hyperparameters, and underlying rationale for these particular selections will be furnished in the revised manuscript.

---

> > ### Comment · Reviewer_XVSh · 2023-08-16
> >
> > I thank the authors for providing clarifications and sharing more results. I suggest the authors to clarify in the paper how consistency relates with robustness in the context of feature attributions [1][2][3]. In the light of the new results, I am happy to raise my score.
> >
> > [1] https://arxiv.org/pdf/2111.00358.pdf
> >
> > [2] https://arxiv.org/abs/1711.00867
> >
> > [3] https://arxiv.org/pdf/1906.07983.pdf

---

### Official Review · Reviewer_k1pA · 2023-07-12

**Soundness:** 4 excellent
**Presentation:** 3 good
**Contribution:** 4 excellent
**Rating:** 6
**Confidence:** 4

**Summary:**

This paper proposes using interpretation error as a metric to gauge inconsistencies and inefficiencies. The authors proposed three algorithms founded on the standard polynomial basis to minimize the interpretation errors. Authors perform experiments using both image and text classification to demonstrate the superior performance in terms of lowest interpretation error to the baseline methods.

**Strengths:**

A major issue for the removal-based explanation is that the change of input distribution with the removal of inputs. This makes the performance evaluation problematic. The trade-off between efficiency and consistency for removal-based explanations is well known. The novelty of this paper is to propose using a single metric, i.e., interpretation error, for evaluating the quality of model interpretation, followed by two algorithms for finding the best interpretation models assuming the interpretable model is polynomial. Authors also conducted comprehensive experiments to evaluate both interpretation error and truthful gap of Harmonica and other baseline algorithms.

**Weaknesses:**

To establish your approach is superior, comparison in terms of prediction accuracy and time complexity across multiple DNN architectures would be helpful. Indeed, the trade-off between prediction performance and interpretability can also be important. It would be nice to see your approach achieves lower interpretation errors without compromising prediction performance.

**Questions:**

Besides efficiency and consistency, there are other factors that would influence the evaluation of model interpretability. Can you also show the prediction accuracy together with interpretability error for your experiments? What about computational complexity?

**Limitations:**

Not really a limitation but rather an assumption: using higher order polynomial to approximate the nonlinearity of DNN.

---

> ### Author Rebuttal · Authors · 2023-08-07
>
> We express our gratitude to the reviewer for dedicating their valuable time and effort to evaluate our manuscript. In the following sections, we provide detailed responses to the reviewer's comments:
>
> **Reply to Weaknesses**
>
> We acknowledge the valid concerns surrounding prediction accuracy and computational efficiency over various DNN architectures. Central to our research is the exploration of interpretation error, providing a lens to assess the efficiency and consistency of removal-based explanations.
>
> While delving deeper into the intricacies of removal-based explanations, such as those typified by LIME and SHAP, we observe that they predominantly yield local or input-specific explanations. The $L_2$ interpretation error metric that we employ can be contextually visualized as a more expansive construct of inefficiency, particularly when the radius converges within 1.
>
> To bolster our argument and cater to the reviewer's concerns, we expanded our experiments across DNN architectures, such as ResNet and VGG. Our additional findings, which can be referenced in **Table 1 and Table 2 in the PDF attached to General Response**, reveal that our method, including Harmonica and its variants, shows their superiority across different settings.
>
> Furthermore, an elucidation of the computational efficiency of these algorithms is shown in the following Table:
>
> **Table 3. Running time of different algorithms on the SST2 dataset using the model mentioned in our paper. The main characteristics affecting the running speed are listed in the table.**
> | | Main characteristics | Running time (s)|
> |---|---|---|
> | Harmonica-2nd  | parallel perturbation generation   | 225|
> | Harmonica-3rd  | parallel perturbation generation  | 616|
> | LIME  | captum.attr.LimeBase (sequential perturbation generation), exponential cosine similarity function   | 1499|
> | IG  | gradient step=500| 82|
> | SHAP  | captum.attr.KernelShap (sequential perturbation generation)   | 1235 |
> | IH  | gradient step=50| 3550|
> | Shapley Taylor Index| generate random perturbations in parallel   | 651|
> | Faith-Shap  | generate random perturbations in parallel   | 538|
>
> It's crucial to highlight, echoing the reviewer's comment, that our algorithms, designed for interpretability, inherently don't affect the prediction accuracy of the original model $f$. In real-world applications, the unaltered model $f$ remains to be utilized for generating predictions, and this paradigm is called the post-hoc explanation. Given our positing of the Impossible Trinity Theorem, the expectation for our algorithm to excel in original efficiency, that is, interpretation error at a radius of 0—which Shapley algorithms pay attention to—may not be theoretically feasible (see **Table 6 in the PDF attached to General Response**). The enriched analyses and insights will seamlessly integrate into our revised manuscript, offering a more comprehensive view of the nuanced trade-offs in play.
>
> **Reply to Questions**
>
> Regarding the questions on other factors influencing the evaluation of model interpretability, we agree with the reviewer's point. Our focus on efficiency and consistency does not exclude the consideration of other essential aspects such as prediction accuracy and computational complexity. In fact, the interpretation error metric can be seen as a generalized version of inefficiency (prediction error), especially when the radius is set to within $1$.
>
> In our additional experiments, we include prediction accuracy (**Table 1 and Table 2 in the PDF attached to General Response**) along with interpretability error as evaluation criteria. This will allow readers to understand the proposed method's performance in a broader context. Additionally, we will provide a thorough analysis of the computational complexity of our algorithms. It is already presented in Appendix C, Theorem 3. This additional information will offer a more in-depth understanding of the computational costs associated with achieving the improvements in interpretation error, thereby providing a more nuanced view of the trade-offs involved in our approach.
>
> **Reply to Limitations**
>
> The assumption regarding the usage of higher-order polynomials to approximate the non-linearity of DNN is indeed an essential aspect of our approach. It is vital to recognize that this assumption while facilitating the development of our algorithms, has theoretical guarantees: all removal-based explanations can be seen as Boolean functions over feature subsets, and all Boolean functions can be represented as Boolean polynomials (using Fourier basis).
>
> However, we have carefully considered this aspect in the design of our algorithms and have chosen low-order polynomial approximation as a reasonable trade-off between model complexity and interpretability. To address the precision issue, we introduce the notion of truthfulness, making this low-order assumption reasonable and quantifiable. In the revised paper, we will expand on this assumption and its implications, providing insights into the rationale behind this choice and its potential limitations. This will further clarify the scope and applicability of our methods.

---

> > ### Comment · Reviewer_k1pA · 2023-08-17
> >
> > Authors rebuttal has satisfactorily addressed my concerns. Thanks.

---

### Author Rebuttal · Authors · 2023-08-09

# General Response

We are deeply grateful for the efforts and valuable feedback of the reviewers and area chairs in reviewing our manuscript. We have made a detailed response to each reviewer's concern, including some clarifications and additional experimental results, and adopted some of the reviewer's suggestions. Some tabular results and figures are placed in attached PDF. We hope these contents can help reviewers understand our manuscript better.

### **More Experimental Results**

In the supplementary PDF, we present more experimental results on:

1. Interpreting different neural network architectures such as ResNet and VGG, interpreting MS-COCO dataset with ground-truth segmentation labels (with illustrative examples).

2. Ablations on the feature subset size when applying the SLIC algorithm for image segmentation.

3. Comparison of different interpretation algorithms on other existing metrics such as Insert/Delection, where Fourier analysis-based method Harmonica still shows consistent superior performance.

We hope these supplementary experiments will illuminate the robustness and efficacy of our method, thereby strengthening both the credibility and relevance of our conclusions.

---

### Decision · Program_Chairs · 2023-09-21

**Decision:**

Accept (poster)

**Comment:**

This paper has had a very positive assessment overall. Overall, I think the authors may have to include a lot of new experimental results and make some changes as suggested by the reviewers. But barring those issues, I recommend acceptance.

The main result is that the authors show an impossibility theorem -- namely, that an interpretable model cannot be simultaneously consistent and efficient for removal-based explanations. The authors first identify that all interpretable models for removal-based explanations can be written as a boolean function over the boolean hypercube formed by a number of variables equal to the number of features. This lets them use Fourier analysis tools.

The authors then show that a sparse polynomial regression model is consistent, at least with respect to the subspace of monomials considered, but is not efficient, while SHAP is efficient but not consistent. Therefore, the authors explore the possibility of a local version of sparse polynomial regression that trades off consistency with efficiency better.

Concerns raised during the rebuttal and discussion were addressed, with the authors performing additional experiments on running time, evaluating removal-based explainers with different baselines, and also expanding experiments for ResNet and VGG architectures.